# Adaptive Elicitation of Latent Information Using Natural Language

**Jimmy Wang** [* 1]  **Thomas Zollo** [* 1]  **Richard Zemel** [1]  **Hongseok Namkoong** [1]

## Abstract

Eliciting information to reduce uncertainty about a latent entity is a critical task in many application domains, e.g., assessing individual student learning outcomes, diagnosing underlying diseases, or learning user preferences. Though natural language is a powerful medium for this purpose, large language models (LLMs) and existing fine-tuning algorithms lack mechanisms for strategically gathering information to refine their own understanding of the latent entity. To harness the generalization power and world knowledge of LLMs in developing effective information gathering strategies, we propose an adaptive elicitation framework that *actively* reduces uncertainty on the latent entity. Since probabilistic modeling of an abstract latent entity is difficult, our framework adopts a predictive view of uncertainty, using a meta-learned language model to simulate future observations and enable scalable uncertainty quantification over complex natural language. Through autoregressive forward simulation, our model quantifies how new questions reduce epistemic uncertainty, enabling the development of sophisticated information gathering strategies to choose the most informative next queries. In experiments on the twenty questions game, dynamic opinion polling, and adaptive student assessment, our method consistently outperforms baselines in identifying critical unknowns and improving downstream predictions, illustrating the promise of strategic information gathering in natural language settings.

## 1. Introduction

The performance of many valuable services and systems depends on the ability to efficiently elicit information and reduce uncertainty about a new environment or problem instance. For example, before an optimal lesson plan can be prepared for a particular student, information must first be gathered about their underlying skills and abilities. Similarly, a patient's health status must be quickly assessed upon intake, while an online service seeking retention aims to gain a fast understanding of a new customer's preferences.

Notably, in these (and many other) cases, the object of interest is *latent*, meaning it cannot be directly measured or observed but can only be queried indirectly. This makes gathering information particularly challenging, as it requires carefully designed strategies to infer the latent entity's characteristics through indirect signals. To achieve efficiency, these strategies must be *adaptive*, dynamically tailoring subsequent queries based on the information gained so far. In the context of student assessment, an adaptive approach might start with broad math questions covering multiple skills. If the student gets a question wrong, the system would then drill down into each relevant skill individually, asking questions of varying difficulty to determine the limits of their proficiency. By progressively refining its queries in this way, the system efficiently maps out the student's knowledge boundaries and thus reduces uncertainty about their individual skill profile (see Figure 6).

As natural language is a particularly powerful and flexible medium for eliciting such latent information, one might assume that modern large language models (LLMs) (Brown et al., 2020; Bai et al., 2022; DeepSeek-AI et al., 2025) could be helpful in such dynamic information gathering efforts. However, LLMs and existing fine-tuning algorithms often treat uncertainty passively, and lack mechanisms for strategically gathering information to refine their own understanding of the latent entity. While existing LLMs are often trained to instill as much *static* world knowledge as possible (Hendrycks et al., 2021), this world knowledge cannot directly be used to reduce uncertainty about *new, unseen* individuals that the model has little information about.

To harness the generalization power and world knowledge of LLMs to address the renewed uncertainty that arises whenever a new environment or individual is encountered, we introduce an *adaptive elicitation framework* that uses natural language to *actively* reduce uncertainty by simulating future responses. Crucially, our approach leverages *meta-learning*,

---

[1]Columbia University. Correspondence to: Jimmy Wang <jw4209@columbia.edu>, Thomas Zollo <tpz2105@columbia.edu>.

*Proceedings of the 42$^{nd}$ International Conference on Machine Learning*, Vancouver, Canada. PMLR 267, 2025. Copyright 2025 by the author(s).

whereby the model is trained on diverse historical question–answer trajectories spanning many latent entities. This meta-trained foundation enables the model to handle new, unseen entities—such as a brand-new student—whose responses are initially unknown and thus create fresh epistemic uncertainty. By aligning a language model's perplexity objective with the goal of predicting all possible (yet unobserved) answers to the questions we might ask, we transform the challenge of directly modeling a latent entity into a simpler and more scalable problem of predicting *masked future observations* (Ye et al., 2024; Fong et al., 2023). As the model observes each new answer from the individual, it systematically *sharpens its beliefs*, distinguishing between uncertainty it can reduce with further data (epistemic) and the inherent noisiness or variability that remains (aleatoric). Our framework enables a wide range of exciting and impactful applications, e.g., constructing a dynamic diagnostic questionnaire that maximizes the information gained about a patient's health or generating a personalized set of test questions that yield the most insight into a student's learning needs (see Figure 1).

**Contribution**   In the remainder of this paper, we introduce our novel framework for latent uncertainty reduction using natural language and demonstrate its effectiveness across several key applications. Our work contributes a key conceptual and algorithmic insight to the accelerating field of LLMs: by obviating the need for directly modeling a distribution over the latent entity and instead employing a predictive view of uncertainty, we enable the development of adaptive information gathering strategies that naturally scale with LLM performance, improving as models become more capable. Our adaptive elicitation framework can be applied directly on top of existing LLMs, enabling the use of internet-scale linguistic knowledge to comprehend uncertainty. Through experiments on tasks such as dynamic opinion polling and adaptive student assessments, we illustrate the versatility and significant potential of our framework to enable more efficient and targeted information elicitation in critical domains and applications. Overall, we aim to lay the foundation for future research into rigorous uncertainty quantification and adaptive decision-making using LLMs, highlighting the promise of active, context-aware strategies in solving real-world problems.

## 2. Adaptive Elicitation Framework

In this section, we present an approach to uncertainty quantification and adaptive question selection in natural language settings where the latent entity cannot be directly modeled. We adopt a *predictive view* of uncertainty: rather than specifying a direct prior or complete model of the latent entity, we focus on how well the model can predict and quantify uncertainty over future observations of that entity. Our method:

(1) **Meta-learns** a predictive language model from historical question–answer data. (2) Uses this model to **quantify uncertainty** about future or unobserved answers from *new, unseen* latent entities, using autoregressive forward simulation to efficiently distinguish between *epistemic* and *aleatoric* uncertainty. (3) Accurately quantifies and sharpens beliefs given new information, and dynamically **adapts question selection** to elicit information that optimally reduces uncertainty. A key advantage of our method is that we can apply our meta-learning method directly on existing pre-trained LLMs, augmenting uncertainty quantification with internet-scale world knowledge.

### 2.1. Problem Formulation

We consider an unobservable latent entity $U \in \mathcal{U}$ (e.g., a student's skill profile or a patient's health status). We query $U$ by posing a question $X \in \mathcal{X}$ (in natural language) and observing an answer $Y \in \mathcal{Y}$ drawn from

$$\text{Answer } Y \sim Q(\cdot \mid \text{Question } X, \text{ Latent } U), \qquad (1)$$

where $Q$ is the ground truth distribution. Our two primary goals are to: (1) *Quantify* our uncertainty about $U$ based on observed question–answer pairs. (2) *Reduce* that uncertainty by adaptively choosing which questions $X$ to ask next.

**Pre-Training**   We assume that we have an abundance of historical data, where a model can learn from *past trajectories* to inform adaptive elicitation about *new, unseen* individuals from which we wish to gather data. We have a collection of entities $U \in \mathcal{U}_{\text{train}}$. For each latent entity $U$ we have access to a sequence of questions and answers $(X_{1:T}^{(U)}, Y_{1:T}^{(U)})$ produced by it. Then our historical pre-training data consists of: $\mathcal{D}_{\text{train}} := \{X_{1:T}^{(U)}, Y_{1:T}^{(U)} : U \in \mathcal{U}_{\text{train}}\}$. For example, an online tutoring service may have an abundance of data about previous students that they may utilize.

**Test-Time Adaptive Selection**   After pre-training, we wish to quantify and reduce uncertainty about new, unseen, latent entities $U_{\text{new}}$ (e.g. a new student or patient). For each $U_{\text{new}}$ we have $T$ rounds where we can sequentially ask questions and receive responses. At each $t = 0, 1, ...T - 1$, the model can adaptively choose a question $X_{t+1} \in \mathcal{X}_{t+1}$ based on previous feedback $\mathcal{H}_t := (X_{1:t}^{U_{\text{new}}}, Y_{1:t}^{U_{\text{new}}})$ and receives an answer $Y_{t+1} \in \mathcal{Y}_{t+1}$.

We will evaluate the model on its ability to predict unobserved answers $Y_{T+1:\infty}$ generated by the latent entity to any future questions $X_{T+1:\infty}$. For example, we may conduct a survey where we can only include 5 questions, but we wish to know the answer to 1000 additional questions. Being able to predict $Y_{T+1:\infty}$ requires the ability to adaptively collect relevant information $Y_{1:T}$, which in turn requires the ability to both quantify and reduce remaining uncertainty about

the entity $U_{\text{new}}$. We detail our approach in the following sections.

## 2.2. Quantifying Uncertainty Using a Predictive Model

Traditional approaches to model uncertainty may try to model $U$ directly (e.g. by assigning a probability distribution over a structured latent space) (Blei et al., 2003; Salakhutdinov & Mnih, 2007). However, specifying such models for complex human-generated responses can be both restrictive and infeasible. For example, it is unclear how to define an explicit parametric model to represent an individual's political opinions. Instead, we adopt a *predictive view* of uncertainty that avoids the need to directly model latent variables. This approach allows us to train autoregressive models directly in the space of natural language, enabling flexible and scalable modeling of its full complexity.

Define the *epistemic uncertainty* to be uncertainty that can be reduced with more information, and the *aleatoric uncertainty* to be uncertainty due to random noise or variation that cannot be reduced by observing more data. Our key observation is that if we were to observe infinite data $Y_{1:\infty}$ produced by the entity, all *epistemic* uncertainty about the entity would disappear. Intuitively, if a teacher could observe a student's answers to a very large set of questions, that teacher could probably completely predict a student's future answers with errors only due to *aleatoric* variation. Examples of this aleatoric uncertainty could include random noise or intrinsic uncertainty in the student's own decision process. This idea is in line with classical views that treat latent variables as unobserved data (Rubin, 1976; Lindley, 1965; Hill, 1968; Dawid, 1984), as well as more modern treatments (Fong et al., 2023; Ye et al., 2024; Zhang et al., 2024).

Under this view, epistemic uncertainty is naturally the uncertainty due to *missing data*: specifically, the uncertainty about unobserved future responses $Y_{t+1:\infty}$ given the current information $Y_{1:t}$. Thus, our objective is to provide accurate uncertainty estimates over missing data $Y_{t+1:\infty}$. In order to quantify uncertainty about $Y_{t+1:\infty}$, we first build off the notion of *entropy*. Given a distribution $P$ with density $p(\cdot)$, the entropy and conditional entropy over an answer $Y \in \mathcal{Y}$ are defined as $H_P(Y) = \sum_{y \in \mathcal{Y}} p(y) \log p(y)$, $H_P(Y \mid \cdot) = \sum_{y \in \mathcal{Y}} p(y \mid \cdot) \log p(y \mid \cdot)$.

Next, let

$$P(Y_{t+1} = y \mid X_{1:t}, Y_{1:t}, X_{t+1} = x) = p_t(y \mid X_{1:t}, Y_{1:t}),$$

which also induces the conditional entropy $H_P(Y_{t+1} \mid X_{1:t}, Y_{1:t})$. Using this notation, we are interested in measuring the uncertainty over missing data

$$\text{Uncertainty (Future Answers} \mid \text{Current Info)} = H_P(Y_{t+1:\infty} \mid X_{1:t}, Y_{1:t}). \quad (2)$$

## Example: Adaptive Student Assessment

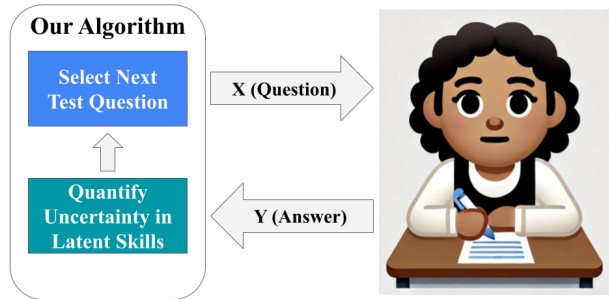

*Figure 1.* Our algorithm can adaptively elicit information from a latent entity via natural language interaction. For example, in assessing a new student, the system may ask questions in areas where the student's abilities are not yet known, to maximize the information gained from each question and efficiently reduce uncertainty about the student's individual skill profile.

Once we have this estimate, we can adaptively choose questions that reduce the greatest amount of uncertainty about the missing data $Y_{t+1:\infty}$ (see Section 2.4 for more details). Notice that this approach to uncertainty quantification works directly in the space of observables $X_{1:\infty}, Y_{1:\infty}$, and does not require any explicit modeling of the latent $U$. If we have a predictive distribution over future answers $Y_{t+1:\infty}$ given previous observations $X_{1:t}, Y_{1:t}$ for every $t$,

$$P(Y_{t:\infty} \mid X_{1:t}, Y_{1:t}) = P(\text{Future Answers} \mid \text{Current Info}),$$

then we can exactly calculate the entropy term in Equation (2). Thus, we directly train an autoregressive predictive model $p_\theta(Y_{t+1:\infty} \mid X_{1:t}, Y_{1:t})$ by optimizing the parameters $\theta \in \Theta$ in order to model these quantities.

## 2.3. Meta-Learning an Autoregressive Predictive Model

To obtain the most accurate estimates of uncertainty, the ideal strategy would be to use the ground-truth answer distribution $Q$ in Equation (1) as the predictive distribution over future observations. In particular, we aim to approximate the conditional distribution $q(Y_{t+1:\infty} \mid X_{1:t}, Y_{1:t})$ induced by Q, which represents the true distribution over future answers given past interactions. Under this setup, the corresponding conditional entropy $H_Q(Y_{t+1:\infty} \mid X_{1:t}, Y_{1:t})$ would yield exact measures of uncertainty. Since $Q$ is unknown in practice, our goal becomes to approximate the conditional $Q$ as closely as possible by training a model $p_\theta$ in order to produce reliable uncertainty estimates.

To approximate $Q$, we assume access to historical data from a collection of latent entities $\mathcal{U}_{\text{train}}$. Each entity $U \in \mathcal{U}_{\text{train}}$ is associated with a sequence of question–answer pairs $\{(X_{1:T}^{(U)}, Y_{1:T}^{(U)})\}$, where $Y \sim Q(\cdot \mid X, U)$ and $U$ is

sampled from a prior distribution. Our first step is to meta-train an autoregressive language model $p_\theta$ on this historical data consisting of sequences of questions and answers from various latent entities

$$\mathcal{D}_{\text{train}} := \{X_{1:T}^{(U)}, Y_{1:T}^{(U)} : U \in \mathcal{U}_{\text{train}}\}.$$

In the student assessment example, $\mathcal{D}_{\text{train}}$ may be a historical dataset of past students, each with an associated sets of test questions and answers. For simplicity, we assume that each sequence is of length $T$, but our framework is agnostic to differing sequence lengths.

**Objective** Define a sequence of previous observations $\mathcal{H}_t := \{X_{1:t}, Y_{1:t}\}$. We train our autoregressive language model $p_\theta$, parameterized by $\theta \in \Theta$, to output one-step probabilities over future answers conditioned on previous observations (i.e., question/answer pairs) $p_\theta(Y_{t+1} | \mathcal{H}_t, X_{t+1} = x)$, inducing a joint distribution over future outcomes

$$p_\theta(Y_{t+1:\infty}|\mathcal{H}_t, X_{t+1} = x_{t+1}, X_{t+2} = x_{t+2}, ...) =$$
$$\prod_{s=t+1}^{\infty} p_\theta(Y_s|\mathcal{H}_{s-1}, X_s = x_s). \quad (3)$$

The training objective for our model is then to optimize the joint log likelihood/marginal likelihood of the observed sequence within the historical dataset

$$\max_{\theta \in \Theta} \left\{ \frac{1}{|\mathcal{U}_{\text{train}}|} \sum_{U \in \mathcal{U}} \sum_{t=1}^{T} \log p_\theta(Y_t^U|\mathcal{H}_{t-1}, X_t^U = x_t) \right\}. \quad (4)$$

After optimizing our model $p_\theta$, we can now use it to approximate the uncertainty estimates $H_Q(Y_{t+1:\infty} \mid X_{1:t}, Y_{1:t}) \approx H_{p_\theta}(Y_{t+1:\infty} \mid X_{1:t}, Y_{1:t})$. To show that optimizing this objective is optimal for approximating $Q$, we first note that maximizing the objective in Equation (4) is equivalent to optimizing an empirical version of the cross entropy $\mathbb{E}_Q[\log p_\theta(Y_{1:T}|X_{1:t})]$. Expanding this loss, we can see that

$$\max_\theta \mathbb{E}_Q[\log p_\theta(Y_{1:T}|X_{1:T})] =$$
$$\max_\theta \{\mathbb{E}_Q[\log q(Y_{1:T}|X_{1:T})] - \mathrm{D}_{\text{KL}}(Q\|p_\theta)\} =$$
$$\mathbb{E}_Q[\log q(Y_{1:T}|X_{1:T})],$$

where $\mathrm{D}_{\text{KL}}(Q\|Q) = 0$. An equivalent interpretation of maximizing the joint log likelihood is that we are minimizing the KL divergence between $p_\theta$ and $Q$, leading to accurate downstream uncertainty estimates. By optimizing this objective over historical data, our aim is for the model to learn meta-learn structures and patterns that will be useful for adaptive testing over *new, unseen* entities.

**Training.** For training, we process each sequence of questions and answers $\{X_{1:N}^{(U)}, Y_{1:N}^{(U)}\}$ corresponding to a latent entity $U$ by sequentially arranging them into one long natural language string $(X_1^{(U)}, Y_1^{(U)}, X_2^{(U)}, Y_2^{(U)}, ...)$. We assume that the probability of a response to a question is independent of the ordering of the earlier questions and answers, and we randomly permute the order of the question/answer pairs within each entity's sequence during training. Then we optimize a language model to predict each answer $Y_t$ conditioned on the current question $X_t$ and previous observations $\mathcal{H}_{t-1}$. To do so, we apply a gradient mask that masks out tokens which do not correspond to any $Y_i$. We use stochastic gradient descent procedures to optimize the training loss.

### 2.4. Adaptive Question Selection by Future Simulation

Having trained a predictive model $p_\theta$ from historical data, we can use this model to quantify uncertainty about future observations generated by the latent entity, and take actions to reduce said uncertainty. Given a new latent entity $U_{\text{new}}$, we may also be interested in different targets of uncertainty: for example, the full sequence of future answers ($Z = Y_{1:\infty}$), a specific subset of questions, or the answer to a particular question ($Z = Y$). To make our notation more general, we notate $Z$ as the object we wish to understand. Our goal is to reduce uncertainty about $Z$ by sequentially choosing questions $X$ that are most informative — i.e., those that will optimally reduce the model's uncertainty about $Z$ (See Figure 6). For example a tutor may be interested in understanding which questions will reveal the most about the student's understanding.

**Setup** As described in Section 2.1, we operate in an adaptive setting at test time. At each round $t = 0, 1, ..., T - 1$, we may select a question $X_{t+1} \in \mathcal{X}_{t+1}$ based on the interaction history so far, $\mathcal{H}_t := (X_{1:t}^{U_{\text{new}}}, Y_{1:t}^{U_{\text{new}}})$, and receive an answer $Y_{t+1} \in \mathcal{Y}_{t+1}$.

To quantify informativeness, we define the information gain from a question-answer pair $(X_{t+1}, Y_{t+1})$ as:

$$\text{IG}_t(Z; (X_{t+1}, Y_{t+1})) =$$
$$H(Z|\mathcal{H}_t) - H(Z|\mathcal{H}_t \cup (X_{t+1}, Y_{t+1})). \quad (5)$$

This measure quantifies the reduction in entropy about $Z$ after observing a new interaction, by quantifying the difference between the current uncertainty $H_{p_\theta}(Z \mid \mathcal{H}_t)$ and the uncertainty after observing $(X_{t+1}, Y_{t+1})$, $H(Z \mid \mathcal{H}_t \cup (X_{t+1}, Y_{t+1}))$. Since we do not yet know $Y_{t+1}$ when choosing $x_{t+1}$, we can instead quantify the *expected* reduction in uncertainty by simulating potential answers using our meta-learned model $p_\theta$. This idea leads to the *Expected*

*Information Gain (EIG) (Chaloner & Verdinelli, 1995)*:

$$\text{EIG}_t(Z; x_{t+1}) =$$
$$H_{p_\theta}(Z \mid \mathcal{H}_t) - \mathbb{E}[H_{p_\theta}(Z \mid \mathcal{H}_t \cup (x_{t+1}, Y_{t+1}))], \quad (6)$$

where we use our meta-learned model $p_\theta(\cdot)$ to simulate $Z$ and $Y_{t+1} \sim p_\theta(\cdot | \mathcal{H}_t, X_{t+1} = x_{t+1})$ in the expectation. To calculate the EIG for multiple choices of $x$, we have

$$\text{EIG}_t(Z; (x_{t+1}, x_{t+2}, ..., x_{t+K})) =$$
$$H_{p_\theta}(Z \mid \mathcal{H}_t) - \mathbb{E}[H_{p_\theta}(Z \mid \mathcal{H}_t \bigcup_{i=t+1}^{t+K} (x_i, Y_i))], \quad (7)$$

where we autoregressively simulate $Y_{t+1:t+K}$ from our meta-learned model. This quantity naturally quantifies the amount of epistemic uncertainty we expect to reduce by choosing a set of questions. If the EIG is very small, then this implies that the reduction in entropy is small and therefore this information is not informative. This could be because there is a lot of aleatoric uncertainty, such that the information gathered is very noisy, or due to the fact that the information gathered is not relevant to the object of interest.

**Question Selection Policies** To select the optimal question at each time $t$, we would ideally like to choose a question selecting policy $\pi : \mathcal{H} \mapsto \mathcal{X}$, where $X_{t+1} \sim \pi(\cdot | \mathcal{H}_t)$, such that we maximize

$$\underset{\pi}{\arg\max} \ \mathbb{E}_{X \sim \pi(\cdot)} \big[\text{EIG}_t(Z; X_{t+1:T})\big], \quad (8)$$

where $\text{EIG}_t(Z; X_{t+1:T})$ is defined in Equation (7). To approximate this quantity, we use our meta-learned model $p_\theta$ to autoregressively simulate possible future answers $Y_{t+1:T} \sim p_\theta$, and the chosen policy $\pi$ to simulate question choices $X_{t+1:T} \sim \pi(\cdot)$. Through autoregressive future simulation, we can optimize for our question selection policy $\pi$. While it is possible to calculate the discrete optimal $x_{t:T}$ that maximizes this objective, it can be intractable as simulating $X_{t+1:T}, Y_{t+1:T}$ is combinatorial in the number of steps. Instead, we introduce two procedures that show strong practical performance while having feasible computational cost.

**Greedy Selection.** A simple question selection policy $\pi^{\text{greedy}}$ is to first enumerate the candidate questions $x \in \{x_1, ..., x_k\}$. Then for each $x_j$, calculate the one-step expected information gain $\text{EIG}_{t:t+1}(Z; x_j)$. Finally, choose the $x_j$ that maximizes this quantity. Concretely,

$$\pi_t^{\text{greedy}} := \underset{x}{\arg\max} \ \text{EIG}_{t:t+1}(Z; x).$$

Although greedy, this policy often performs well in practice and is computationally simpler than globally optimal planning. We establish the theoretical validity of this procedure in Section B, where Proposition B.3 bounds the performance gap between a full combinatorial planning approach and the greedy selection procedure.

**Lookahead / Monte Carlo Planning.** To account for multi-step effects (e.g., a question that might not immediately reduce much uncertainty but paves the way for more informative follow-ups) and to better approximate the combinatorial quantity in Equation (8), we can apply standard *Monte Carlo Tree Search* (MCTS) techniques from reinforcement learning (Browne et al., 2012; Silver et al., 2016). We use a simple instantiation with strong empirical performance and leave more complex variants to future exploration.

With an MCTS policy $\pi^{\text{MCTS}}$, we sample entire simulated question–answer sequences using the meta-learned model $p_\theta$ up to depth $d$ to estimate the cumulative information gain. In order to simulate future responses, we use $\pi_{\text{greedy}}$ to select questions and the Information Gain (5) as proxy rewards. Starting at time $t$, we first calculate $\text{EIG}_{t:\infty}(Z; X_{t+1:\infty})$ for each $x \in \mathcal{X}_t$, and choose the top $K$ questions. For each of the $K$ questions, we then simulate $N$ futures up to depth $d$. For each sample path $i \in [N]$, we receive reward $r^{(i)}(x) = \text{IG}_t\left(Z; (X_{t+1:t+d}^{(i)}, Y_{t+1:t+d}^{(i)})\right)$, where questions are sequentially selected using $\pi^{\text{greedy}}$ and answers are simulated using the meta-learned model $p_\theta$. Finally, the MCTS policy chooses an action as

$$\pi^{\text{MCTS}} := \underset{x \in \mathcal{X}_t}{\arg\max} \ \frac{1}{N} \sum_{i=1}^{N} \text{IG}_t\left(Z; (X_{t+1:t+d}^{(i)}, Y_{t+1:t+d}^{(i)})\right).$$

Though more expensive computationally, we find that $\pi^{\text{MCTS}}$ can find better long-horizon query strategies.

## 3. Experiments

To rigorously benchmark adaptive information gathering strategies for LLMs, we require datasets that (i) capture diverse latent entities or hidden factors, (ii) provide many possible queries about these entities, and (iii) for each entity, link some queries to corresponding ground-truth answers. Such an experimental setup allows us to assess the ability of an LLM and/or particular algorithm to strategically select questions in order to reduce uncertainty about the latent entity. Ideally, each dataset reflects the real-world complexities of human-generated responses, while still providing enough structure for robust evaluation of different query selection policies. In practice, a large pool of possible questions with many ground truth answers is essential, since it allows an adaptive strategy to actively and deeply explore the latent entity along many dimensions, while still leaving unobserved data for evaluation. Then, each latent entity (e.g., a survey respondent's political stance, a stu-

dent's hidden skill profile, or the identity of a secret object) can be progressively unveiled by observing how it answers newly selected queries. Such design criteria enable controlled, quantitative evaluations of LLMs under interactive, information gathering scenarios.

Our experiments focus on three applications: the "Twenty Questions" game (using our novel and publicly available dataset, described below), opinion polling, and student assessment. In each scenario, the objective is to adaptively select questions that reveal as much information as possible with respect to a separate (though potentially overlapping) set of target questions. Questions are chosen one at a time, and each new question–answer pair is appended to the LLM's context before proceeding. For every experiment, we start with a dataset containing a collection of latent entities $U$, each associated with a set of questions $X^U$ and answers $Y^U$. To train our meta-learned model $p_\theta$, we split each dataset by groups of latent entities into training, validation, and test sets. We first meta-learn $p_\theta$ on question–answer pairs corresponding to the training entities, after which we evaluate how effectively the model quantifies and reduces epistemic uncertainty about observations generated by test entities. Further details regarding the datasets, training procedure, baseline comparisons, and evaluation metrics are provided below.

### 3.1. Twenty Questions Dataset

The classic "Twenty Questions" game epitomizes our core goal of reducing uncertainty about a hidden entity through targeted queries. Specifically, the object (such as an animal, a musical instrument, or any of a wide range of other concepts) serves as the latent factor $U$ that cannot be directly observed. A player (or model) must identify $U$ by posing a sequence of questions and observing the corresponding answers (e.g., "no," "maybe," or "yes"). Hence, the game inherently captures the essence of characterizing a latent entity by uncovering how it generates answer to different possible questions. To operationalize this game for benchmarking, we construct a novel "Twenty Questions" dataset from a curated set of objects in the THINGS database ([Hebart et al., 2019](#)), each serving as a potential hidden entity. For each object, we produce a diverse set of candidate questions (*e.g.*, "Does it have four legs?", "Is it edible?", "Is it used for entertainment?") together with the corresponding answers, generated by a top-quality LLM (Claude-3.5-Sonnet). In total, the dataset contains 800 objects, each with answers to a set of 1200 questions. By treating each object as a distinct latent entity, we capture a broad spectrum of scenarios, ranging from everyday items ("banana," "telephone") to more uncommon concepts ("violin," "canoe"). We note that the absolute correctness of Claude's answers is not crucial, as our goal is for our model to learn the underlying data-generating process governing which answers appear,

given specific questions and objects. Our dataset is publicly available,[1] including the complete set of objects, curated questions, generated answers, and relevant metadata.

### 3.2. Other Datasets

**OpinionQA ([Santurkar et al., 2023](#))** Originally created to evaluate the alignment of LLM opinions to those of 60 US demographic groups, this dataset contains 1498 multiple choice political questions answered by a diverse collection of survey respondents. These questions target various political issues ranging from abortion to automation. For each question $X$, the multiple choice answer corresponds to the observable feedback $Y$, and the survey respondent's latent political preference corresponds to the unobservable $U$.

**EEDI Tutoring Dataset ([Wang et al., 2020](#))** EEDI is an online educational and tutoring platform that serves millions of students around the globe. This dataset includes a collection of 938 math questions focusing on various areas such as algebra, number theory, and geometry, as well as individual responses from many students. Each question is a multiple choice question with four answers that includes a visual diagram as well as associated text. The student's true mathematical ability $U$ generates the student's answer $Y$ to the math question $X$.

### 3.3. Meta-Training Details

We first split the training datasets by entity into train, validation, and test with a 70%, 15%, 15% split. To meta-train our model, we initialize a pre-trained Llama-3.1-8B model in FP16 precision and use LoRA ([Hu et al., 2021](#)) to finetune our model with parameters $\alpha = 24$, rank$= 8$, and dropout$= 0.1$. Additional details are shown in Appendix [C.1](#)

### 3.4. Baselines

Here we describe the baselines to which we compare our algorithm; each consists of an approach to model fine-tuning, and an approach to question selection. As in our method, each chosen question–answer pair is appended to the LLM context for predicting unseen answers.

**Base LLM** First, we consider a simple baseline. For an LLM we use Llama-3.1-8B, from which our meta-trained model is initialized, with no additional fine-tuning; question selection is performed randomly.

**In-Context Tuning (ICT)** Next, we consider a typical in-context learning (ICL) baseline. First, we meta-train the

---

[1] https://huggingface.co/datasets/
namkoong-lab/TwentyQuestions

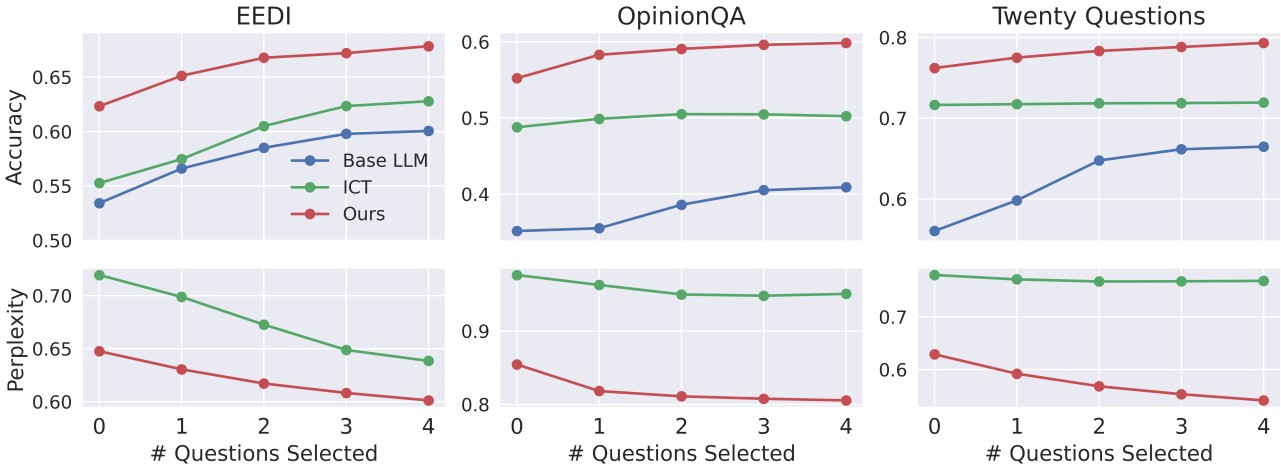

*Figure 2.* Accuracy (top) and perplexity (bottom) of our adaptive elicitation framework compared to baseline methods across three datasets: OpinionQA, EEDI student assessment, and Twenty Questions. The x-axis represents the number of questions selected. Our method works best to gather information and accurately characterize the latent in each case. Each plot is the average of 10,000 simulations across unseen entities.

model via In-Context-Tuning (Chen et al., 2022), where the objective is to predict the label for a query example given some number of in-context support examples. Then, questions are selected based on embedding similarity to the target questions that we aim to answer (Liu et al., 2021). We use the same model and parameters as described in Section 3.3, and we use Alibaba-NLP/gte-large-en-v1.5 as our embedding model.

### 3.5. Evaluation

To evaluate how well each method can ask targeted questions to reduce uncertainty about the latent entity, we perform 10,000 trials, where on each trial we randomly select an entity and apply our algorithm (and baselines). For each trial and its corresponding entity, we randomly select a pool of $N$ **candidate questions** from which the methods can sequentially choose questions to ask, and randomly select $K$ held-out **target questions**. The objective is to sequentially choose optimal questions from the candidate questions to reduce the most uncertainty about the held-out target questions for the entity. In our experiments, we choose $N = 20$ and $K = 5$, but we include ablations that vary these quantities in Section 4.5. We evaluate the performance on the target questions with four metrics: (1) Accuracy, (2) Perplexity (Jelinek et al., 1977), (3) Expected Calibration Error (Guo et al., 2017), and (4) Brier Score (Brier, 1950).

## 4. Results and Discussion

In this section, we empirically study the following questions: (1) Can our framework be used to adaptively select questions to reduce uncertainty and elicit information about the latent entity? (2) Do we generate reasonable posterior probability updates and reduce uncertainty as more information is gathered? (3) When is this adaptive procedure particularly helpful, and when is advanced planning (i.e., MCTS) most important? (4) How crucial is our training procedure for producing actionable uncertainty quantification? (5) Does the performance of our framework improve with a better underlying LLM? Throughout, we connect these findings to the paper's broader motivation: the importance of adaptive strategies to eliciting information efficiently in real-world scenarios.

### 4.1. Overall Gains from Adaptive Elicitation

Overall results for our method and 2 baselines across all 3 datasets are shown in Figure 2. The top row of plots record accuracy on the target questions, while the bottom row record perplexity (or negative log-likelihood loss). The Base LLM is omitted on bottom for ease of visualization. Our framework is applied using the greedy EIG strategy. In both figures, the X-axis records the number of questions that have been selected so far.

Across all 3 datasets and both metrics, our algorithm most effectively characterizes the latent entity by predicting the answers to target questions (we show similar results for Brier Score in Figure 7). Further, our algorithm consistently improves its characterization as more information is gathered, whereas gathering more questions based on embedding distance does not always help. Overall, our adaptive elicitation framework proves effective in gathering information and reducing uncertainty across 3 diverse domains.

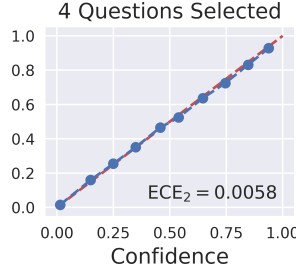
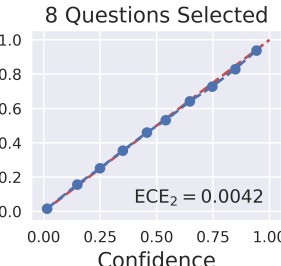

Figure 3. Reliability diagrams comparing confidence and accuracy after different numbers of selected questions (and observed answers). Our model maintains well-calibrated uncertainty estimates, increasing both confidence and accuracy as more questions are asked.

## 4.2. Uncertainty Quantification

A cornerstone of our approach is using *predictive* perplexity as an indicator of uncertainty to guide the adaptive strategy; this makes sense only if our model's probabilities correctly reflect confidence about unseen data. To assess this, we examine calibration, or the extent to which the model's confidence reflects its prediction accuracy.

For each dataset, we plot reliability diagrams (Guo et al., 2017) of confidence vs. accuracy, where perfect calibration lies on the $y = x$ line, and record Expected Calibration Error (ECE). Both the reliability diagram and ECE are produced by separating predictions into 10 bins by confidence, and comparing the average confidence and accuracy for each bin. Results are shown after 1, 4, and 8 questions are selected, and the far left subfigure displays overall average confidence and accuracy for each setting.

Results for OpinionQA are shown in Figure 3, while EEDI and Twenty Questions are shown in Appendix Figures 8 and 9. For all 3 datasets, we observe that the predicted probabilities lie close to the diagonal of perfect calibration—our model's confidence aligns well with actual accuracy. As more questions are observed, the model's average confidence (and accuracy) both go up, confirming that uncertainty diminishes in an intuitive way. In the motivating student-assessment scenario, this means that by asking just a few strategically chosen questions, the model not only improves its predictions but also becomes *more certain* in them. For a high-stakes application such as medical diagnostics or skill placement exams, it is crucial to know when a model has enough data to be sure in its predictions, versus when it is still uncertain; these calibration results confirm our framework performs well in this sense.

## 4.3. When is Adaptivity Most Helpful?

Having established that our adaptive question selection method is generally effective at quantifying uncertainty and eliciting information about some latent, we next examine *when* such a procedure is most helpful. In particular, we

hypothesize that adaptive strategies are especially important in characterizing features of the latent which are relatively rare in the population. As a concrete example, while many students may have overlapping weaknesses (e.g., many get the same test question wrong), it can be harder to learn that a particular student is struggling in an area where other students generally do not. An adaptive strategy could help by selecting a test question that most find easy but this student may answer incorrectly.

To investigate this hypothesis, we specify two different subgroups of questions as targets by running an evaluation where for each target question in the subgroup, the entity's answer must have probability less than either 50% ("medium") or 30% ("hard") across the population. We use our meta-trained model with random, EIG, and MCTS question selection, and record results after N questions have been selected. Results are shown in Figure 4. For each question subgroup (as well as all questions from the previous experiment), we record on the y-axis the relative accuracy gain from using EIG or MCTS, compared to random selection.

First, we notice that the more advanced MCTS planning strategy outperforms EIG in all cases, and both always outperform random. Intuitively, while a greedy strategy picks the single next question that locally maximizes immediate information gain, it may miss questions whose short-term yield seems small but that pave the way for far more revealing follow-up queries. By looking multiple steps ahead, MCTS better accounts for how each query reshapes future options, often enabling it to find more globally optimal questioning strategies. This means that given a good model for uncertainty quantification, we can improve our results by spending more compute, indicating good scaling behavior in our algorithm.

Next, we observe trends across different subgroups of questions. In all 3 example applications, adaptivity and planning have a massive impact on the ability to answer hard questions compared to random question selection. For EEDI and Twenty Questions the percent gain over random with

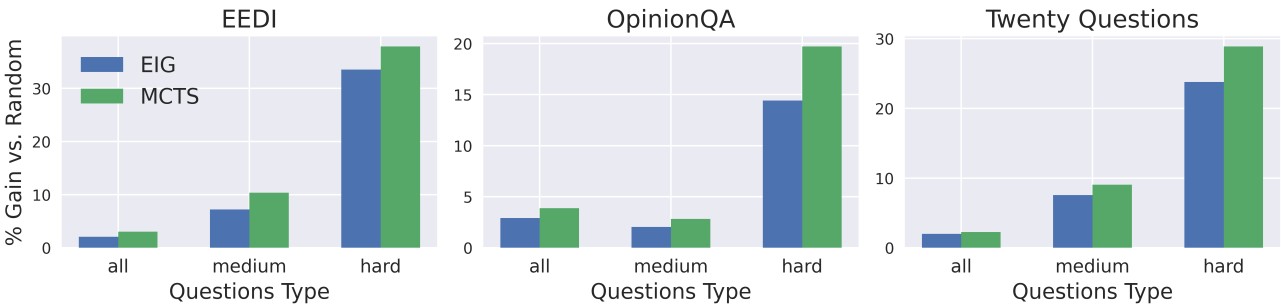

Figure 4. Relative accuracy gain from adaptive question selection (EIG and MCTS) over random selection for different subsets of target questions: all, medium difficulty (answer observed in $< 50\%$ of the population), and hard (answer observed in $< 30\%$). Adaptivity provides the greatest benefits when identifying rare latent traits, demonstrating when strategic question selection is most advantageous.

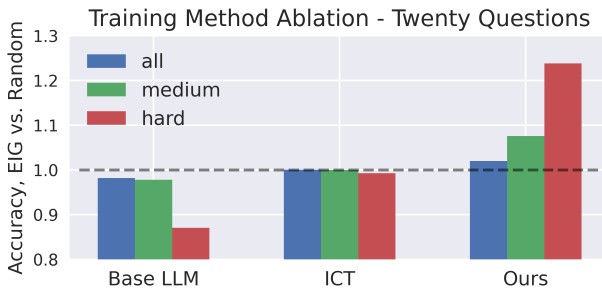

Figure 5. Comparison of performance gains from planning (EIG-based selection) using different models.

EIG or MCTS is more than 10x higher for hard questions than for all questions; for OpinionQA, it is 5x higher. We thus have strong evidence that our adaptive information elicitation strategy is most important when characterizing the latent features which are most atypical with respect to the population. If the latent entity exhibits atypical behavior (a student struggles with a concept that most find easy, or an opinion respondent holds a rare viewpoint), an adaptive method can target precisely those concepts that discriminate such cases. Conversely, random or fixed questionnaires fail to unearth those nuances within a limited query budget.

### 4.4. Training Ablation

Our results in Figure 2 confirm the effectiveness of our end-to-end adaptive elicitation framework, while Figure 4 demonstrates the significant gains from planning-based question selection over random selection. Now, we turn to understanding the remaining component—meta-training—by evaluating how planning performs when applied to our model versus the ICT model and base LLM. We use the Twenty Questions dataset, and the same splits of all, medium, and hard questions as the previous experiment. For each setting and each of 3 underlying models, (Base, ICT, and ours), we record the accuracy on target questions after

selecting 3 questions with either random selection or the EIG strategy. To measure what is gained from planning, we record the ratio of target question accuracy with planning to that with random selection (a value above 1 indicates some accuracy gain from planning).

Results are shown in Figure 5. First, we see that planning performs poorly using the Base LLM, reducing accuracy almost 15% on hard questions compared to random question selection. The ICT model performance is largely unchanged by planning, across all 3 question types. On the other hand, our model's performance is greatly improved when question selection is guided by planning, highlighting that our training procedure is essential to enable such strategic information gathering with LLMs.

### 4.5. Other Ablations

We first ablate the number of candidate and target questions to choose from. Our experiments were run with the models being able to select from 20 questions in order to accurately predict 5 targets. In Table 2 in Appendix E, we find that our method gains more accuracy as the question bank becomes larger. In Table 1, we find that performance stays roughly the same as the number of target questions changes. Finally, we study the effect of the base model for our meta-training procedure. We test GPT2, Llama-3.2-1B, and Llama-3.1-8B, and find in Table 3 that performance increases as the model is larger.

## 5. Conclusion

In this work, we propose an adaptive elicitation framework, based on the missing-data view, that actively reduces uncertainty on the latent entity by simulating counterfactual responses. There is a rich body of literature on topics related to latent entity modeling and its applications, of which we are only able to give a limited overview here. See Appendix A for a thorough discussion of related research.

## Acknowledgements

We thank ONR Grant N00014-23-1-2436 for its generous support. This work is supported by the funds provided by the National Science Foundation and by DoD OUSD (R&E) under Cooperative Agreement PHY-2229929 (The NSF AI Institute for Artificial and Natural Intelligence).

## Impact Statement

The broader impact of this work includes significant potential for enhancing human-centered systems and services that require adaptive decision-making based on incomplete information. Our approach enables more targeted and personalized interactions, which could improve educational tools, assistive technologies, and personalized healthcare assessments. However, the ethical implications of such a framework must be carefully considered. Adaptive elicitation systems can shape user experiences, influence decision-making, and reinforce biases present in training data. We acknowledge the importance of ongoing scrutiny to ensure that such systems are deployed responsibly, particularly in high-stakes applications like healthcare and education.

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

Figure 6. An example of how adaptive elicitation (bottom) can improve over static strategies (top) in new student assessment. Each question asked to the student is marked green (answered correctly) or red (answered incorrectly). Once a student answers incorrectly, the adaptive strategy is able to search over the ***high-dimensional question space***, and present the student with a series of more granular examples that resolve uncertainty about specific abilities in the ***high-dimensional space representing the student's latent abilities***. In this case, active question selection reveals that the student is strong in division, but struggles with decimal points. A static assessment, on the other hand, fails to resolve this uncertainty.

## Reproducibility

Our code is available at `https://github.com/namkoong-lab/adaptive-elicitation`.

## A. Related Work

**Latent Uncertainty Modeling and Decision Making**     Traditional works that attempt to model latent uncertainty to make robust decisions often pose explicit Bayesian models that directly specify a latent parameter. Classical multi-armed bandits such as Thompson Sampling-based methods (Russo, 2020; Grover et al., 2018; Chapelle & Li, 2011; Agrawal & Goyal, 2013; Lattimore & Szepesvári, 2019; Jun et al., 2016; Li et al., 2010) specify a Bayesian model as a prior, which can be used to draw explicit posterior samples. Examples of Bayesian models include Gaussian and Bernoulli distributions, as well as Bayesian linear or logistic regression (Russo et al., 2020). Bayesian Optimal Experimental Design (BOED) methods (Ghavamzadeh et al., 2015; Chaloner & Verdinelli, 1995; Ryan et al., 2016) follow a similar paradigm by quantifying information gain using explicit Bayesian models to make sequential decisions.

Other lines of work include more sophisticated probabilistic modeling. Active collaborative filtering methods (Boutilier et al., 2003) specify probabilistic models of user preference data. Bayesian Optimization techniques (Frazier et al., 2008; Frazier, 2018; Gonzalez et al., 2016; Jiang et al., 2020; Jones et al., 1998) apply acquisition functions such as Expected Improvement (Jones et al., 1998) and Knowledge Gradient (Frazier et al., 2008) on top of Gaussian Processes. While these traditional methods are statistically principled, the need to specify explicit models means they often struggle to model very high-dimensional spaces such as the space of natural language. For example, (Frazier, 2018) mentions that Bayesian Optimization methods are best suited to dimensionality $\leq 20$, while natural language embeddings are often thousands of dimensions.

To overcome these limitations, recent works have focused on representing uncertainty over high dimensions augmented by the representation power of neural networks. One line of works explicitly uses neural network representations to represent the underlying uncertainty, where decision making algorithms are applied on top of Bayesian Neural Networks (BNNs) or the last layer representation (Snoek et al., 2015; Riquelme et al., 2018; Osband et al., 2016; 2018; Piech et al., 2015). Another strand of works uses ensembles to represent the underlying uncertainty (Qin et al., 2022), or more efficient variants such as Epistemic Neural Networks (Wen et al., 2021; Osband et al., 2022; 2023b). We offer a different approach by directly modeling the uncertainty surrounding future predictions using language models. Methods that explicitly represent uncertainty in terms of ensembles or through specified parameters still struggle to operate in the discrete, high-dimensional space of natural language, whereas our perspective is able to directly represent uncertainty in natural language predictions. Additionally, our meta-learning method does not require new architectures and can be applied on top of powerful pre-trained language models, allowing the use of internet-scale language understanding in comprehending uncertainty.

**Computerized Adaptive Testing** Our work is closely related to Computerized Adaptive Testing (CAT) methods, a form of educational testing that adapts to a student's ability level. Classical CAT methods attempt to capture a student's latent ability level through simple parametrized models. Item Response Theory, also known as Latent Response Theory, includes a family of simple mathematical models such as a one parameter logistic regression or Item Response Function to model a student's latent ability (Liu et al., 2024; Embretson & Reise, 2013). The Diagnostic Classification Model (DCM) is designed to measure proficiency across a wide array of specific knowledge concepts. A prominent example includes the DINA method (Torre, 2009; de la Torre, 2011) which uses a probabilistic binary matrix model to represent these concepts. Knowledge Tracing techniques, which train machine learning methods to model the latent knowledge of students as they interact with coursework, traditionally use Hidden Markov Models (HMMs) (Corbett & Anderson, 1994) or Partially Observable Markov Decision Processes (POMDPs) (Rafferty et al., 2011) to model the latent state. More modern treatments use deep learning models such as a Recurrent Neural Network (Piech et al., 2015) to represent latent knowledge.

**Reinforcement Learning with Sequence Models** A number of works propose to train or use powerful pre-trained models in order to solve complex reinforcement learning (RL) tasks, focusing on how these models can make decisions using vast amounts of offline data (Janner et al., 2021; Yang et al., 2023; Chen et al., 2021; Du et al., 2024; Lee et al., 2022). Another line of works show that using meta-learned sequence models to predict the next action can approximate standard bandit algorithms (Lin et al., 2024a; Lee et al., 2023; Zhang et al., 2024). We extend these ideas to natural language while focusing on how our meta-learned model can quantify uncertainty to make a decision.

**Uncertainty Quantification over Natural Language.** There has been a recent class of works focusing on developing uncertainty measures to augment the reliability of model responses. (Kuhn et al., 2023; Lin et al., 2024b; Malinin & Gales, 2021; Duan et al., 2024) focus on predictive entropy measures with off-the-shelf language models, while other approaches focus on self-consistency in the generation space (Lin et al., 2022; Si et al., 2023; Kadavath et al., 2022; Diao et al., 2023). Another class of works focuses instead on detecting *epistemic* uncertainty from *aleatoric* uncertainty in model outputs (Yadkori et al., 2024; Osband et al., 2023a; Hou et al., 2024; Glushkova et al., 2021). Our meta-learning uncertainty quantification framework is complementary to these works, as these measures are designed to be applied on top of pre-trained foundation models.

**Planning and Information Gathering with LLMs** Our work is related to Uncertainty of Thoughts (UoT) (Hu et al., 2024) and OPEN (Handa et al., 2024). While these methods build elicitation procedures on top of off-the-shelf language models, we use a meta-learning procedure in order to accurately quantify uncertainty over new environments. Other works introduce methods to enhance general reasoning or planning capabilities by using natural language reasoning steps (Wei et al., 2022; Wang et al., 2022; Yao et al., 2023).

**Personalization with Language Models**   With the recent success of Large Language Models (LLMs), a natural question is whether these models can be tailored and personalize to various users. There has been a nascent series of works that propose benchmarks and methods for personalized language modeling. (Zollo et al., 2025; Kirk et al., 2024; Castricato et al., 2024) propose new testbeds and evaluation criteria that target various dimensions of personalization through synthetic and real data. (Jang et al., 2023) proposes to merge model parameters to personalize models, while (Li et al., 2024b; Poddar et al., 2024) propose new personalized fine-tuning and Reinforcement Learning from Human Feedback (RLHF) techniques. In the context of opinion polling, (Li et al., 2024a) steers model outputs to different personas by using embeddings from collaborative filtering, while (Park et al., 2024) demonstrates that language models can successfully be adapted to individual responses.

## B. Theoretical Validity

We now detail two propositions that provide insight into our meta-learning framework. The first proposition lower bounds performance of our model in terms of the performance in the ideal simulation as well as the difference between the true and simulated distribution. The second proposition quantifies the performance gap between a greedy and full combinatorial method, showing the losses one might expect to incur.

First, let $\mathcal{X}_T = (x_1, ..., x_T)$. Define $\mathcal{X}^*$ to be the optimal set of questions $X$ that maximizes the log likelihood of the object of interest $Z$ under the meta-learned distribution $p_\theta$,

$$\mathcal{X}^* := \underset{\mathcal{X}_T}{\text{argmax}} \ \mathbb{E}_{p_\theta}[\log p_\theta(Z \mid \mathcal{X}_T)].$$

This is the set of questions we would ask if we could perfectly optimize in our simulated environment. Next, let $q$ be the true distribution and $p_\theta$ our meta-learned model. Using these definitions, we state our first proposition

**Proposition B.1.** *For any $p_\theta$,*

$$\mathbb{E}_q[\log p_\theta(Z \mid \mathcal{X}^*)] \geq \mathbb{E}_{p_\theta}[\log p_\theta(Z \mid \mathcal{X}^*)] - \sqrt{\mathbb{E}_{p_\theta}[\log^2 p_\theta(Z \mid \mathcal{X}^*)]\chi^2(q(Z) \parallel p_\theta(Z \mid \mathcal{X}^*))}. \tag{9}$$

$\mathbb{E}_q[\log p_\theta(Z \mid \mathcal{X}^*)]$ represents the cross entropy of $Z$ between $p_\theta$ and the ground truth distribution $q$. The lower bound first involves $\mathbb{E}_{p_\theta}[\log p_\theta(Z \mid \mathcal{X}^*)]$, which is the likelihood of the data under our simulated distribution. The second term involves both the likelihood under the simulated distribution $\mathbb{E}_{p_\theta}[\log^2 p_\theta(Z \mid \mathcal{X}^*)]$ as well as the distance between $q$ and $p$ through $\chi^2(q(Z) \parallel p_\theta(Z \mid \mathcal{X}^*))$. This bound tells us that if $p_\theta$ has high likelihood under the simulator and has little distance to $q$, then we are guaranteed to achieve good test-time performance. However, since the difference $\chi^2$ is scaled by the likelihood, simply having high likelihood in the simulated distribution is not enough. In fact, if $\chi^2(q\|p)$ is large, then having high likelihood in the simulation can exacerbate this error in the second term. We prove this in Appendix B.2.

**Bound for Greedy Policy**   Next, we bound the performance gap between the questions collected from the optimal policy $\mathcal{X}^*$ with those collected from the greedy policy

$$x_i^{\text{greedy}} = \underset{x_i}{\text{argmax}} \, \mathbb{E}_{p_\theta}[\log p_\theta(Z \mid (x_1, ...x_{i-1}) \cup x_i)],$$

where $\mathcal{X}_{\text{greedy}} = (x_1^{\text{greedy}}, ..., x_T^{\text{greedy}})$. First we define the concept of submodularity:

**Definition B.2** (Submodularity). *$f : 2^\Omega \to \mathbb{R}$ is submodular if $\forall X \subseteq Y \subseteq \Omega$ and $\forall z \notin Y$ we have*

$$f(X \cup \{z\}) - f(X) \geq f(Y \cup \{z\}) - f(Y)$$

Then we state our proposition:

**Proposition B.3.** *Under the assumption that the entropy over $Z$ produced by the meta-learned model $p_\theta$ is submodular,*

$$\mathbb{E}_{p_\theta}[\log p_\theta(Z \mid \mathcal{X}^*)] - \mathbb{E}_{p_\theta}[\log p_\theta(Z \mid \mathcal{X}_{greedy})] \leq \frac{1}{e}\text{EIG}(Z; \mathcal{X}^*).$$

We prove this statement in Appendix B.3. This proposition states that the difference in achieving the optimal log likelihood under the simulated environment and using the greedy strategy is bounded by $\frac{1}{e}\text{EIG}(Z; \mathcal{X}^*)$. We can then use this bound

and substitute in Proposition B.1 to quantify the performance lower bound for the greedy policy. In this proposition, we have to assume submodularity of our meta-learned model because it is not guaranteed in practice due to training instabilities or errors. Empirically we find that the entropy of our meta-learned model behaves submodularly, as evidenced by the perplexity graphs in Figure 2.

### B.1. Summary

Our framework provides a data-driven, natural-language-based alternative to parametric modeling of latent entities. It proceeds by: (1) *Meta-training* a language model on diverse question–answer sequences, (2) Interpreting the model's predictive distribution over future answers as an *uncertainty measure* about new entities, and (3) Iteratively *selecting questions* to optimally reduce that uncertainty. We show theoretically that our procedure gives strong performance, even under a simple and efficient greedy planning strategy. Next, we explore the empirical performance of our framework in a series of adaptive information gathering scenarios.

In this section, we show the theoretical validity of using a greedy procedure to select actions with the highest expected information gain. We wish to quantify and reduce our uncertainty about some object $Z$ by choosing the optimal questions $X$ to query the latent entity $U$. First we define the expected information gain of asking a set of questions $\mathcal{X}_t := (x_1, ..., x_t) \subseteq \mathcal{X}$,

$$\text{EIG}(Z; \mathcal{X}_t) \; = \; H(Z) \; - \; \mathbb{E}_t\big[H\big(Z \mid \bigcup_{s=1}^{t}(x_s, Y_s)\big)\big].$$

Note that each $Y_s$ in the history is a random variable, and we use our meta-learned model $p_\theta$ to simulate possible answers $Y_s \sim p_\theta(\cdot | \mathcal{H}_{s-1}, X_s = x_s)$. Similarly, $Z \sim p_\theta(\cdot)$ as well. Ultimately, our goal is to choose a set of designs $\mathcal{X}_t = x_{1:t}$ that yields the largest amount of information gain possible. First to set notation, define $q$ to be the ground truth underlying question and answer distribution. Let $p$ be the distribution induced by the meta-learned model, and $p(Z|\mathcal{X}_t)$ be the conditional distribution over $Z$ after marginalizing out the feedback $Y_{1:t}$ corresponding to the questions $\mathcal{X}_t$, where $Y_{1:t}$ comes from the ground truth distribution such that $Y_{1:t} \sim q(\cdot)$.

First, define $\mathcal{X}^*$ to be the optimal set of questions $X$ that maximizes the log likelihood of the object of interest $Z$ under the meta-learned distribution $P$,

$$\mathcal{X}^* := \underset{\mathcal{X}_t}{\operatorname{argmax}} \; \mathbb{E}_p[\log p(Z \mid \mathcal{X}_t)].$$

### B.2. Proof of Proposition B.1

We restate our proposition for clarity.

$$\mathbb{E}_q[\log p(Z \mid \mathcal{X}^*)] \geq \mathbb{E}_p[\log p(Z \mid \mathcal{X}^*)] - \sqrt{\mathbb{E}_p[\log^2 p(Z \mid \mathcal{X}^*)]\chi^2(q(Z) \parallel p(Z \mid \mathcal{X}^*))}.$$

*Proof.* We can decompose the difference between the cross entropy term $\mathbb{E}_q[\log p(Z \mid \mathcal{X}^*)]$ and entropy term $\mathbb{E}_p[\log p(Z \mid \mathcal{X}^*)]$ as

$$
\begin{aligned}
& - \mathbb{E}_q[\log p(Z \mid \mathcal{X}^*)] + \mathbb{E}_p[\log p(Z \mid \mathcal{X}^*)] \\
&= \int (p(z|\mathcal{X}^*) - q(z)) \log p(z \mid \mathcal{X}^*) dz \\
&= \int (1 - \frac{q(z)}{p(z \mid \mathcal{X}^*)}) \log p(z \mid \mathcal{X}^*) p(z \mid \mathcal{X}^*) dz \\
&\leq \sqrt{\left(\int (1 - \frac{q(z)}{p(z \mid \mathcal{X}^*)})^2 p(z \mid \mathcal{X}^*) dz\right) \left(\int \log^2(p(z \mid \mathcal{X}^*)) p(z \mid \mathcal{X}^*) dz\right)} \\
&= \sqrt{\mathbb{E}_p[\log^2 p(Z \mid \mathcal{X}^*)]\chi^2(q(Z) \parallel p(Z \mid \mathcal{X}^*))}.
\end{aligned}
$$

where the second to last line follows from the Cauchy-Schwartz inequality and the last line follows from the definition of the $\chi^2$ divergence. Then by flipping the sign, we obtain the stated inequality. $\square$

## B.3. Proof of Proposition B.3

We first define submodularity for clarity:

**Definition B.4** (Submodularity). $f : 2^\Omega \to \mathbb{R}$ is *submodular* if $\forall X \subseteq Y \subseteq \Omega$ and $\forall z \notin Y$ we have

$$f(X \cup \{z\}) - f(X) \geq f(Y \cup \{z\}) - f(Y)$$

In order to show that the greedy procedure is able to perform close to the optimal solution, we rely on the following assumption, that the entropy calculated from our meta-learned model is submodular:

**Assumption B.5** (Submodularity of Entropy). Let $\mathcal{H}_t \subseteq \mathcal{H}'_t$. Then for any $(X_{t+1}, Y_{t+1}) \notin \mathcal{H}'_t$,

$$H(Z \mid \mathcal{H}_t \cup (X_{t+1}, Y_{t+1})) - H(Z \mid \mathcal{H}_t) \geq H(Z \mid \mathcal{H}'_t \cup (X_{t+1}, Y_{t+1})) - H(Z \mid \mathcal{H}'_t).$$

We first show that the Expected Information Gain (EIG) is submodular. If the entropy is submodular, then the Information Gain is also submodular. Define the Information Gain as

$$f(\mathcal{H}_t) = \text{IG}(Z; \mathcal{H}_t) = H(Z) - H(Z \mid \mathcal{H}_t).$$

Then, for any history set $\mathcal{H}_t$ and any additional observation $(X_{t+1}, Y_{t+1}) \notin \mathcal{H}_t$, the marginal gain of adding $(X_{t+1}, Y_{t+1})$ is given by

$$\begin{aligned}
f(\mathcal{H}_t \cup \{(X_{t+1}, Y_{t+1})\}) - f(\mathcal{H}_t) &= \Big[ H(Z) - H(Z \mid \mathcal{H}_t \cup \{(X_{t+1}, Y_{t+1})\}) \Big] - \Big[ H(Z) - H(Z \mid \mathcal{H}_t) \Big] \\
&= H(Z \mid \mathcal{H}_t) - H(Z \mid \mathcal{H}_t \cup \{(X_{t+1}, Y_{t+1})\}).
\end{aligned}$$

Now, consider two history sets $\mathcal{H}_t \subseteq \mathcal{H}'_t$ and the same observation $(X_{t+1}, Y_{t+1}) \notin \mathcal{H}'_t$. By our submodularity assumption on the entropy, we have

$$H(Z \mid \mathcal{H}_t) - H(Z \mid \mathcal{H}_t \cup \{(X_{t+1}, Y_{t+1})\}) \geq H(Z \mid \mathcal{H}'_t) - H(Z \mid \mathcal{H}'_t \cup \{(X_{t+1}, Y_{t+1})\}).$$

In terms of the Information Gain function, this inequality becomes

$$f(\mathcal{H}_t \cup \{(X_{t+1}, Y_{t+1})\}) - f(\mathcal{H}_t) \geq f(\mathcal{H}'_t \cup \{(X_{t+1}, Y_{t+1})\}) - f(\mathcal{H}'_t).$$

Thus, by definition the Information Gain is submodular. Since this is true for all $X_{t+1}, Y_{t+1}$, then the Expected Information Gain (EIG) is also submodular. Then by (Nemhauser et al., 1978), we have that

$$\text{EIG}(Z; \mathcal{X}_{\text{greedy}}) \geq (1 - \frac{1}{e}) \text{EIG}(Z; \mathcal{X}^*),$$

implying that

$$\text{EIG}(Z; \mathcal{X}_{\text{greedy}}) - \text{EIG}(Z; \mathcal{X}^*) \leq \frac{1}{e} \text{EIG}(Z; \mathcal{X}^*). \tag{10}$$

Finally, to prove the bound we can note that

$$\begin{aligned}
\mathbb{E}_p[\log p(Z \mid \mathcal{X}^*)] - \mathbb{E}_p[\log p(Z \mid \mathcal{X}_{\text{greedy}})] &= \mathbb{E}_Y[\mathbb{E}_Z[\log p(Z \mid \mathcal{X}^*, \mathcal{Y}^*))]] - \mathbb{E}_Y[\mathbb{E}_Z[\log p(Z \mid \mathcal{X}_{\text{greedy}}, \mathcal{Y}_{\text{greedy}})]] \\
&= \mathbb{E}_Y[H(Z \mid \mathcal{X}^*, \mathcal{Y}^*)] - \mathbb{E}_Y[H(Z \mid \mathcal{X}_{\text{greedy}}, \mathcal{Y}_{\text{greedy}})] \\
&= \mathbb{E}_Y[H(Z \mid \mathcal{X}^*, \mathcal{Y}^*)] - H(Z) + H(Z) - \mathbb{E}_Y[H(Z \mid \mathcal{X}_{\text{greedy}}, \mathcal{Y}_{\text{greedy}})] \\
&= \text{EIG}(Z; \mathcal{X}_{\text{greedy}}) - \text{EIG}(Z; \mathcal{X}^*) \\
&\leq \frac{1}{e} \text{EIG}(Z; \mathcal{X}^*),
\end{aligned}$$

where the last line follows from an application of Equation (10). This completes the proof.

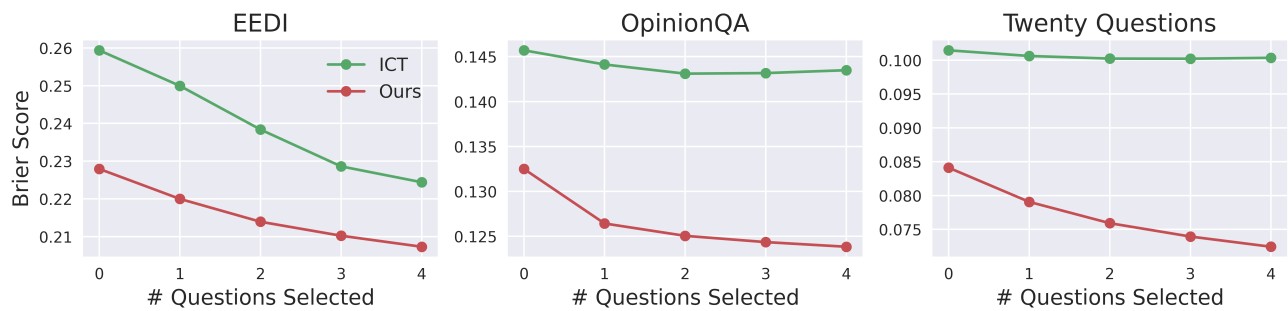

*Figure 7.* Brier score results in our overall setting across 3 datasets.

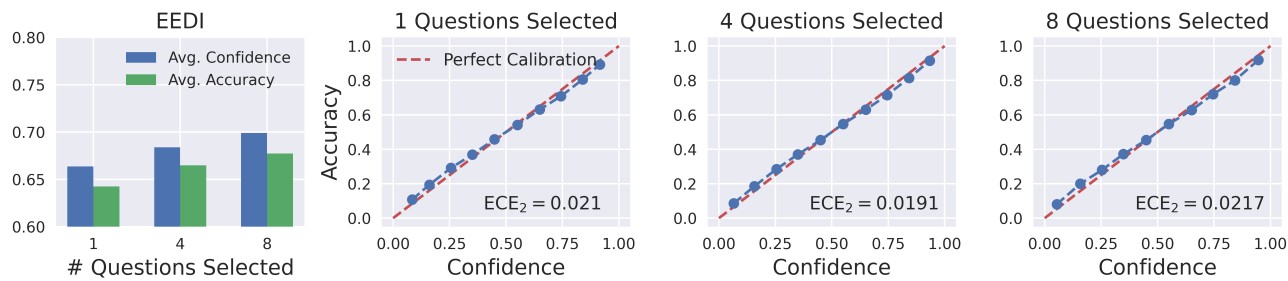

*Figure 8.* Calibration results with EEDI.

## C. Experiment Details

### C.1. Meta-Training Details

We initialize the AdamW (Loshchilov & Hutter, 2019) optimizer with learning rate of $0.0001$ and $\beta = (0.9, 0.95)$, weight decay of $0.1$, and we use a linear warmup for the learning rate after which we use a cosine scheduler. We train our model for $10,000$ epochs with a batch size of $4$ and block size of $1024$, after which we take the checkpoint with the lowest validation loss.

## D. Experiment Results

Here we include additional experiment results. Figure 7 shows results for the overall experiments with the Brier Score metric. Figure 8 shows calibration results for EEDI, and Figure 9 shows calibration results for Twenty Questions.

## E. Ablations

Additional ablations are shown in Table 1 (number of target questions), Table 2 (number of candidate questions), Table 3 (base model).

*Table 1.* Ablating number of target questions on EEDI; accuracy conditioned on 4 Questions

| Accuracy | **1** | **5** | **10** | **20** |
|---|---|---|---|---|
| Base | 0.6042 | 0.6005 | 0.6066 | 0.5987 |
| Ictx | 0.6269 | 0.6278 | 0.6295 | 0.6255 |
| Ours | **0.6759** | **0.6784** | **0.6871** | **0.6832** |

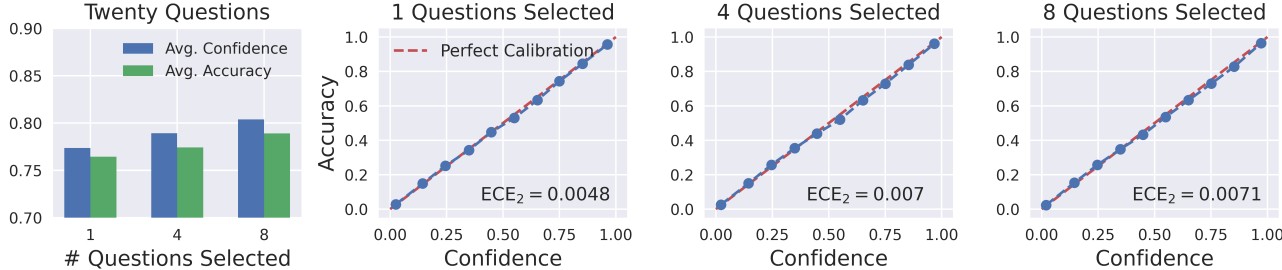

*Figure 9.* Calibration results with Twenty Questions.

*Table 2.* Ablating number of candidate questions on OpinionQA; accuracy conditioned on 4 Questions

| Accuracy | 10 | 15 | 20 | 25 |
|---|---|---|---|---|
| Base | 0.4030 | 0.4042 | 0.4089 | 0.4093 |
| Ictx | 0.4988 | 0.4993 | 0.5023 | 0.5009 |
| Ours | 0.**5933** | **0.5953** | **0.5987** | **0.6068** |

*Table 3.* Ablating base model: Twenty Questions accuracy using our framework conditioned on 4 questions

| | GPT2 | Llama-3.2-1B | Llama-3.1-8B |
|---|---|---|---|
| Accuracy | 0.5201 | 0.6131 | 0.7382 |

