# OpenReview forum: "Adaptive Elicitation of Latent Information Using Natural Language"
_ICML.cc/2025/Conference — ICML 2025 poster_

### Official Review · Reviewer_54sR · 2025-03-11

**Overall Recommendation:** 4

**Summary:**

This paper studies the problem of adaptively selecting queries to reduce uncertainty on a latent entity. They propose to fine-tune an LLM for meta-learning the task of question answering with a latent entity (e.g. the 20 questions game with different hidden entities). This approach allows for measuring uncertainty of the latent entity as the LLM’s predictive uncertainty in predicting answers to future questions. This approach is shown to be effective in producing calibrated uncertainty estimates which allows for better uncertainty-guided question selection.

**Claims And Evidence:**

The claim that the adaptive query selection strategy is better than a random selection strategy is supported by results in Figure 3 as well as Figure 5.

The claim that the meta-learning method effectively learns to model uncertainty is supported by Figure 4 where the ECE is clearly very low. It’s unclear though how exactly the confidence is calculated in this case. It is stated in Section 2.3 that the “variability in these simulated futures” is treated as a measure of uncertainty, but it is not mentioned anywhere what measure of variability is used and how uncertainty is converted to confidence. I believe some notion of entropy is used, but more details about how the entropy is computed would help clarify this point.

The claim that their method works for general latent information elicitation settings is supported by their experiments across three diverse applications.

**Essential References Not Discussed:**

The Uncertainty of Thoughts paper is mentioned in the related work, but it is unclear why it is not directly compared against in any of the experiments.

**Experimental Designs Or Analyses:**

The experiment in Figure 3 and the analysis for how well the proposed method can perform adaptive elicitation is valid. The examination of calibration to determine effectiveness of their measure of uncertainty is also a valid approach. Their further analysis on how well their method performs on the most challenging questions is very valuable and provides useful insight into where their approach works best. Finally, the ablation on the underlying model in their framework is a valid way to show that the meta-training method is vital to their method’s success.

**Methods And Evaluation Criteria:**

The proposed methods and evaluation metrics make sense.

**Other Comments Or Suggestions:**

- Line 36 left: The sentence is either missing punctuation or generally does not make sense.
- Line 402 right: “setti” -> “setting”
- Line 652 is missing a closing parenthesis.

**Other Strengths And Weaknesses:**

Other strengths:
- The problem considered is well motivated as highly significant.
- The experiments and results are presented in a clear manner.

Other weaknesses:
- The experiments seem to be lacking on baselines. It is well supported that the proposed LLM fine-tuning method is useful for uncertainty quantification, and the selection method is useful for the question selection task, but it is unclear how this approach fairs compared to the other methods which address these problems.
- A discussion on the computational expense of the method would be helpful since it seems like if there are thousands of candidate questions, then even the greedy method could be very slow.

**Questions For Authors:**

1. Why are existing LLM uncertainty quantification methods not comparable baselines?
2. How is confidence, as shown in Figure 4, calculated?

**Relation To Broader Scientific Literature:**

The method presented in this paper for fine-tuning an LLM for the task of meta-learning next action prediction has been studied in prior work in reinforcement learning, but this paper uses the idea for LLM uncertainty quantification. The problem setting in this paper has also been studied in prior work such as Uncertainty of Thoughts which similarly tries to determine the best questions to ask to elicit the most information, but this paper shows that fine-tuning an LLM for meta-learning is actually useful for this setting.

**Theoretical Claims:**

I looked over the proof of Theorem A.4 which bounds the performance gap of the greedy approach and an optimal planning method. The proof is relatively straightforward, although I did not look into all of the details.

---

> ### Author Rebuttal · Authors · 2025-03-31
>
> We thank the reviewer for their thoughtful feedback.  We are encouraged that the reviewer found the problem setting to be well motivated, and our experiments and results to be presented clearly.  Please see our responses to the particular concerns raised.
>
>
> **[Q1: Calculating confidence]**
>
> Confidence is calculated in the typical way as the softmax response for the predicted answer.  In response to this concern, we will clarify this calculation in our revised paper, as well as clearly specify when we use related notions of entropy or variability.
>
> **[Q2: Computational Expense]**
> In terms of analysis, we provide a brief overview for the reviewer. Let $n$ be the number of possible questions to ask and $m$ be the number of possible answers to each question. The computational complexity of the EIG method at each time step $t$ is $O(m\cdot n)$. This is because we need to calculate the conditional entropy of each answer for each feasible question in the set. To calculate the complexity of MCTS, let $z$ be the number of trajectories we simulate and $d$ be the depth of simulation. Then MCTS is $O(z\cdot n\cdot m\cdot d) $, as we have $z$ simulations up to depth $d$ for each question, and at each depth we need to perform an EIG calculation. We will provide a detailed computational complexity analysis in our final draft.
>
> To compare the complexity of our method to other approaches,  we note that other common methods like embedding-based methods will require a linear scan of the possible questions, yielding a complexity similar to our greedy method. Furthermore, methods that directly use LLMs to generate questions and responses require forward passes in the number of tokens in each question and responses, while our method only requires one forward pass in this respect. Applying MCTS on top of using LLMs to generate responses (as in UoT) will incur complexity $O(z\cdot n\cdot m\cdot d)$ multiplied by complexity of generating responses.
>
> We acknowledge the complexity of the MCTS approach can be a bottleneck in scenarios with an extremely large question space or stringent real-time constraints. However, we also want to highlight that there are also many important applications (including some that we consider) where this should not be prohibitive.
>
> For example, in educational settings the extra computation can be performed during the student’s response time, as the policy is updated while the student works on the current question—thus introducing a tolerable lag. Conversely, in real-time applications such as live diagnostics or interactive dialogue systems, the computational overhead of MCTS might be prohibitive, and more efficient strategies (e.g., greedy selection) may be preferred. We view our work as introducing a new conceptual perspective on uncertainty-guided questioning, and we hope that future work will build on our approach to develop more computationally efficient solutions.
>
>
>
> **[Q3: Comparison to other methods]**
>
> While there exists a growing body of literature on UQ in LLMs, much of this work is focused on different settings and types of uncertainty than we address here.  In particular, much of this growing body of work is focused on short-form generation and QA style tasks [e.g., 1-5].  These works are primarily focused on improving the reliability in off-the-shelf LLMs, by quantifying existing uncertainty in some answer to a response to some question. This is a fundamentally different task than quantifying how observing the answer to some question will affect your uncertainty in other question/answers pairs, and using these estimates to make adaptive decisions to reduce uncertainty.
>
> With respect to Uncertainty of Thoughts (UoT), we did not compare as a baseline because UoT uses the LLM to *generate* potential questions, while our setting involves choosing potential questions from a given question bank. If we adapt UoT to our setting, it would be equivalent to applying the EIG measure to the base LLM. We compare this in the ablation in Figure 6, and show that adapted UoT (EIG + base LLM) performs worse than random selection. We hypothesize that this worse performance is due to the specialized nature of datasets like OpinionQA and EEDI which may not be in the training set of the base LLM, pointing to the necessity of our meta-learning procedure.
>
> - [1] Semantic Uncertainty: Linguistic Invariances for Uncertainty Estimation in Natural Language Generation	https://arxiv.org/abs/2302.09664
> - [2] Shifting Attention to Relevance: Towards the Predictive Uncertainty Quantification of Free-Form Large Language Models	https://arxiv.org/abs/2307.01379
> - [3] Language Models (Mostly) Know What They Know	https://arxiv.org/abs/2207.05221
> - [4] Just Ask for Calibration: Strategies for Eliciting Calibrated Confidence Scores from Language Models Fine-Tuned with Human Feedback	https://arxiv.org/abs/2305.14975
> - [5] INSIDE: LLMs' Internal States Retain the Power of Hallucination Detection	https://arxiv.org/abs/2402.03744

---

### Official Review · Reviewer_hSjM · 2025-03-14

**Overall Recommendation:** 3

**Summary:**

Authors consider a meta-learning QA scenario in which each dataset contains an unobservable latent variable (e.g., medical notes with an unseen clinical diagnosis). They propose an iterative and adaptive framework designed to reduce uncertainty around these latent variables.

**Claims And Evidence:**

**Claim 1.** "We introduce an adaptive elicitation framework that uses natural language to actively reduce uncertainty on the latent entity by simulating counterfactual responses"

Yes, the framework is introduced in Sec. 2. I am not sure that the responses are actually counterfactual in this framework. See “Questions For Authors”.

**Claim 2.** "Our approach enables the model to identify epistemic uncertainty and facilitate sophisticated information gathering strategies as it updates its understanding of the latent ..."

Even though the method provides some form of uncertainty over a latent entity, I don’t see clear justification that predictive perplexity estimates epistemic uncertainty over a latent entity.

**Claim 3**. Experimentally, “we illustrate the versatility and significant potential of our framework to enable more efficient and targeted information elicitation in critical domains and applications.”

This claim is well-supported by the experiments.

**Essential References Not Discussed:**

I think more relevant works should be mentioned in related work section, for instance:

### Planning with LLMs
Language Models as Zero-Shot Planners: Extracting Actionable Knowledge for Embodied Agents https://arxiv.org/pdf/2201.07207

ReAct: Synergizing Reasoning and Acting in Language Models https://arxiv.org/pdf/2210.03629

### UQ for LLMs
Kernel Language Entropy: Fine-grained Uncertainty Quantification for LLMs from Semantic Similarities https://arxiv.org/abs/2405.20003

Reducing Conversational Agents’ Overconfidence Through Linguistic Calibration https://aclanthology.org/2022.tacl-1.50/

**Experimental Designs Or Analyses:**

I think the datasets and baselines are well-selected. The only issue I see is the lack of computational efficiency analysis.

**Methods And Evaluation Criteria:**

The methods and evaluation criteria are logical. However, incorporating a realistic use case would make the argument more compelling.

**Other Comments Or Suggestions:**

Section 2: the method explanation in the beginning was unclear,

The exchangeability assumption is overly strong and known to be incorrect based on findings from cognitive psychology, such as order effects, framing, and anchoring effects. It is important to discuss and emphasize that this assumption is not realistic.

It would be useful to provide examples of questions and answers in the paper, and show how the uncertainty changes in practice.

**Other Strengths And Weaknesses:**

### Strength:
- Original work and important topic,
- The method is valid and supported by the experimental results

### Weakness:
- Some assumptions (exchangeability) are not well discussed, and are not realistic
- The method in my opinion could be motivated better, e.g.,
    * Why optimizing the joint log likelihood/marginal likelihood is optimal?
    * Why is simulation needed to quantify uncertainty?

**Questions For Authors:**

I'm puzzled by the term "counterfactual response" in the paper. Could you clarify its meaning? I don't see how the responses generated by your framework qualify as counterfactual. Are you using "counterfactual" in a different sense than its common usage in causal inference literature?

**Relation To Broader Scientific Literature:**

I think the paper discusses well the relation to broader scientific literature. The work nicely expands works on LLM planning and uncertainty quantification to a an important application of reducing uncertainty over a latent entity in a dialogue.

**Theoretical Claims:**

1. The assumption of exchangeability is too strong. See “Other Comments and Suggestions” for further details.

2. The proof of A4 was difficult to follow. Specifically:
- Steps (5-7) require more detailed explanations.
- The definition and role of fact A2 are unclear.
- The bound $\log⁡(∣Y∣(T−t))log(∣Y∣(T−t))$ (LL680) is not well-explained.
- It is unclear why assumption A2 is referred to as a fact.

As a result, I was only able to fully understand the bound of the third term.

---

> ### Author Rebuttal · Authors · 2025-03-31
>
> We thank the reviewer for their thoughtful and thorough review, and their recognition that our claims are well-supported by experiments. We response below to the concerns.
>
> **[Q1: Use of "counterfactual response"]**
>
> We apologize for the confusing use of the term "counterfactual response" in the paper. "Counterfactual" has no reference to its usage in the causal inference literature and a better term to use would be "simulated response". We will replace this in the final draft.
>
> **[Q2: Predictive perplexity / Joint likelihood]**
>
> Our formulation of uncertainty quantification derives from a large body of work on inference with missing data (e.g. [1-2]).  Under this view, knowing $U$ is defined as being able to predict any answer $Y$, with errors only due to random, aleatoric variation. Observing the infinite sequence $(X_{1:\infty}, Y_{1:\infty})$ reduces all epistemic uncertainty, leaving only aleatoric uncertainty: $H(Y \mid U) = H(Y \mid X_{1:\infty}, Y_{1:\infty})$. Then, epistemic uncertainty in $U$ naturally corresponds to uncertainty in the *missing data* $Y_{t+1:\infty}$ given history $Y_{1:t}$. The joint likelihood $P(Y_{t+1:\infty} \mid X_{1:t}, Y_{1:t})$ exactly quantifies uncertainty about the missing data $Y_{t+1:\infty}$ which shows it is the correct objective. Under the special condition of exchangeability, this is precise: there exists $\theta(X_{1:\infty}, Y_{1:\infty})$, such that $H(Y \mid U) = H(Y \mid \theta(X_{1:\infty}, Y_{1:\infty}))$.
>
> We will include a detailed description in the paper.
>
> **[Q3: Simulation to quantify epistemic/aleatoric?]**
>
> Simulating future trajectories allows us to quantify which actions will reduce the most epistemic uncertainty. To quantify uncertainty about a target $Z$ we calculate,
>     $$\text{Epistemic Uncertainty}(Z | X_{1:t}, Y_{1:t}) = (\text{Total}) H(Z \mid X_{1:t}, Y_{1:t}) - (\text{Aleatoric}) \mathbb{E}[H(Z \mid X_{1:\infty}. Y_{1:\infty})].$$ In practice, we approximate this quantity using simulation to calculate existing epistemic/aleatoric uncertainty. The EIG then quantifies the reduction in epistemic uncertainty, which we also calculate by simulation.
>
> **[Q4: Exchangability assumption]**
> We agree with the reviewer that the exchangeability assumption may be too strong in practice. We will outline these fundamental limitations in our camera-ready version. We also emphasize that exchangeability mainly serves as inspiration for our practical UQ framework, alongside our strong empirical results.
>
> **[Q5: Computational analysis]**
> Define $n$ as the number of questions to ask and $m$ be the number of possible answers.  The computational complexity of the EIG method at each time step $t$ is $O(m\cdot n)$ because we need to calculate the conditional entropy of each answer for each feasible question in the set. The complexity of MCTS is $O(z\cdot n\cdot m\cdot d)$, where $z$ is the number of simulations and $d$ be the depth of simulation, since there are $z\cdot d$ EIG calculations. We will include a detailed discussion of complexity analysis in the camera-ready version.
>
> **[Q6: Proof of A4]**
>
> Steps (5-7) in the proof outline a telescoping expansion of the KL divergence term. The telescoping sum consists of three terms. (5) is the difference between the ground truth distribution $q$ and $p$ conditioned on the optimal information $X_{p_{\theta}}^{\ast}$. (6) is the difference of $p(Y_{t:T} | X_{p_{\theta}}^{*})$ when $Y_{t:T}$ is simulated from $q$ versus $p$. Finally (7) measures the difference of conditioning on the optimal information $X_{p_{\theta}}^{\ast}$ and $X_{greedy}$.
>
> Regarding Fact A.2, we apologize as we made a typo. Fact A.2 refers to the fact that for any $S \in \mathcal{S}, 0\leq H(S) \leq \log |S|$ on lines 671-674. The term $\log (|Y| (T-t))$ comes from bounding $H(Y_{t:T} | X_{p_{\theta}}^{*})$ on line 685. Because $Y_{t:T}$ is exchangeable and the ordering does not matter, then the cardinality of $Y_{t:T}$ is $|Y|(t-T)$.
>
> **[Q7: Realistic use cases]**
> Thank you for this suggestion, we agree that incorporating more realistic use cases could strengthen the evaluation, especially in high-impact areas like medical diagnosis.  While 20 Questions is synthetic, we highlight that EEDI and OpinionQA are real-world datasets in personalized tutoring and opinion polling, important domains where a technique like ours might make significant impact.
>
> **[Q8: Related work/qualitative examples]**
> We thank the reviewer for pointing out missing related work; we will include these and additional references in our final version. We will also include examples of questions and answers in the paper, and show how uncertainty changes in practice.
>
> [1] Edwin Fong, Chris Holmes, and Stephen G Walker. Martingale posterior distributions. Journal of the Royal Statistical Society, Series B, 2023.
>
> [2] Naimeng Ye and Hongseok Namkoong. Exchangeable sequence models quantify uncertainty over latent concepts. arXiv preprint arXiv:2408.03307, 2024.

---

### Official Review · Reviewer_nfGQ · 2025-03-14

**Overall Recommendation:** 4

**Summary:**

This paper introduces an adaptive elicitation framework for actively reducing uncertainty about latent entities, using adaptive query selection and simulated counterfactual responses. It leverages a meta-trained LLM to quantify uncertainty about future or unobserved answers via simulation, then iteratively selects queries that maximize expected information gain. Experiments on dynamic opinion polling, adaptive student assessment, and a structured 20 Questions game demonstrate that the method significantly improves accuracy and uncertainty reduction.

**Claims And Evidence:**

The claims made in the submission are supported by clear and convincing evidence, including empirical results across three different tasks (Opinion QA, EEDI tutoring, and 20 questions), and the theoretical analysis on the expected information gain.

**Essential References Not Discussed:**

To my knowledge, there are no essential related works that are missed.

**Experimental Designs Or Analyses:**

* Validity of the experimental design is checked.

The baselines (in-context tuning, base LLM with random question selection) use the same backbone model or meta-training setting as in the main experiment, which is a fair comparison. The split is done with latent entities instead of just QA pairs, which avoids data leakage. A consistent improvement in performance and a reduction in uncertainty are clear across all three datasets and evaluation metrics, showing the effectiveness of the adaptive elicitation framework. The ablation studies are extensive, including comparisons between adaptive (EIG, MCTS) and random selection for different subsets of difficulty, the comparisons of performance gains from planning using different models, and the comparisons of different number of target questions.

**Methods And Evaluation Criteria:**

The use of LLMs meta-trained with historical data containing latent entities is reasonable and well-motivated. Predictive perplexity as a measure of uncertainty makes sense, as it provides a practical and scalable way to approximate epistemic uncertainty without requiring explicit latent variable modeling. For the evaluation, the selected datasets cover diverse domains, and the authors select common metrics for uncertainty quantification, such as ECE and Brier score.

**Other Comments Or Suggestions:**

* It would be helpful if the authors could outline "the rich body of literature on topics related to latent entity modeling" in the appendix.

**Other Strengths And Weaknesses:**

Strengths:

* Uncertainty-aware decision-making in LLMs is an important topic, and the specific focus of eliciting latent entities with meta-training and adaptive question selection appears a novel contribution.

* The selected datasets are from various domains, and effectively demonstrate the framework’s potential for latent entity elicitation.

* Code is available

* The paper is well-presented and easy to follow

Weaknesses:

* Lacking analysis and comparison on the overall complexity of the method (especially for variants with MCTS)

**Questions For Authors:**

* Is the performance sensitive to the choice of the embedding model? Have you tried such ablation with different embedding models?

**Relation To Broader Scientific Literature:**

The paper builds on previous RL research demonstrating that pretrained models can be adapted for decision-making tasks with offline data, and meta-learned sequence models approximate bandit algorithms, but instead focuses on the natural language setting and uncertainty quantification in decision making. It is complimentary and aligns with the works on uncertainty quantification over natural language. It also fit into the category of planning and information gathering with LLMs,. However, unlike prior methods that rely on off-the-shelf models for uncertainty estimation, it uses a meta-learning procedure.

**Theoretical Claims:**

I did not check the proofs in the Appendix due to time constraint.

---

> ### Author Rebuttal · Authors · 2025-03-31
>
> We thank the reviewer for their effort in reviewing our paper, and their thoughtful feedback.  We are pleased that the reviewer appreciated the importance of the topic as well as the novelty of our approach.  Below please see our responses to the particular concerns raised.
>
> **[Q1: Sensitivity to embedding model]**
> As per the reviewer's suggestion, we ran additional experiments to ablate the importance of the embedding model. We tested 5 embedding models: stella_en_1.5B_v5, Qwen2-7B-instruct, e5-mistral-7b-instruct , OpenAI text-embedding-3-large, and AlibabaNLP/gte-large-en-v1.5 (which we used in our original paper), and measured their accuracy after conditioning on 3 in-context questions for the EEDI and OpinionQA dataset. We set the number of targets to be 5 and number of possible questions to ask as 20, which mirrors the setup in our paper. Here are the experiment results, where numbers reported are for each entry is an average over 10,000 different runs:
>
> | Embedding Model                         | EEDI Accuracy | OpinionQA Accuracy |
> |----------------------------------------|---------------|--------------------|
> | AlibabaNLP/gte-large-en-v1.5           | 0.621         | 0.505              |
> | stella_en_1.5B_v5                      | 0.618         | 0.510              |
> | gte-Qwen2-7B-instruct                  | 0.625         | 0.501              |
> | e5-mistral-7b-instruct                 | 0.626         | 0.506              |
> | OpenAI text-embedding-3-large          | 0.631         | 0.515              |
>
> We find that while there is a bit of variation in the embedding model performance. For EEDI, we notice that the larger 7B models and OpenAI text-embedding-3-large have slightly higher performance. For OpinionQA, the performance is more varied. However, none of the embedding models make the In-Context Tuning + Embedding baseline outperform our method. Our method achieves 0.678 accuracy on EEDI and 0.597 accuracy on OpinionQA, as reported in Figure 3.
>
> **[Q2: Computational Complexity]**
>
> In terms of analysis, let $n$ be the number of possible questions to ask and $m$ be the number of possible answers to each question. The computational complexity of the EIG method at each time step $t$ is $O(m\cdot n)$. This is because we need to calculate the conditional entropy of each answer for each feasible question in the set. To calculate the complexity of MCTS, let $z$ be the number of trajectories we simulate and $d$ be the depth of simulation. Then MCTS is $O(z\cdot n\cdot m\cdot d) $, as we have $z$ simulations up to depth $d$ for each question, and at each depth we need to perform an EIG calculation. We will provide a detailed computational complexity analysis in our final draft.
>
> To compare the complexity of our method to other approaches,  we note that other common methods like embedding-based methods will require a linear scan of the possible questions, yielding a complexity similar to our greedy method. Furthermore, methods that directly use LLMs to generate questions and responses require forward passes in the number of tokens in each question and responses, while our method only requires one forward pass in this respect. Applying techniques like MCTS on top of using LLMs to generate responses will incur complexity $O(z\cdot n\cdot m\cdot d)$ multipled by complexity of generating responses.
>
> We acknowledge that the complexity of the MCTS approach can be a bottleneck in scenarios with an extremely large question space or stringent real-time constraints. However, we also want to highlight that there are also many important applications (including some that we consider) where this should not be prohibitive.
>
> For example, in educational settings the extra computation can be performed during the student’s response time, as the policy is updated while the student works on the current question—thus introducing a tolerable lag. Conversely, in real-time applications such as live diagnostics or interactive dialogue systems, the computational overhead of MCTS might be prohibitive, and more efficient strategies (e.g., greedy selection) may be preferred. We view our work as introducing a new conceptual perspective on uncertainty-guided questioning, and we hope that future work will build on our approach to develop more computationally efficient solutions (e.g., via distillation).
>
> In response to the reviewer's concern, we will include a more detailed discussion of these trade-offs and complexity considerations in the camera-ready version.
>
> **[Q3: Related work on latent entity modeling]**
>
> We agree with the reviewer that a review surrounding latent entity modeling would be of major benefit. We will make sure to include this discussion in the camera-ready version of the paper.

---

> > ### Comment · Reviewer_nfGQ · 2025-04-03
> >
> > Thanks for the response and additional experiments on embedding models, which addressed my concerns. I have raised my score to 4.

---

### Official Review · Reviewer_BfZB · 2025-03-20

**Overall Recommendation:** 3

**Summary:**

This paper introduces a framework for adaptive elicitation of latent information using LLMs to optimize question selection based on predictive uncertainty. Instead of explicitly modeling latent variables (e.g., knowledge levels or user preferences), the method quantifies uncertainty via LLM perplexity and simulated future responses to guide information gain-based question selection. The approach, tested on opinion polling, student assessment, and the Twenty Questions game, outperforms baselines by efficiently reducing uncertainty. The key contributions include uncertainty-driven question selection (via Greedy EIG and MCTS) and a general-purpose adaptive questioning framework applicable to diverse domains like education and healthcare.

**Claims And Evidence:**

The claim can be supported by the paper

**Essential References Not Discussed:**

None

**Experimental Designs Or Analyses:**

The method relies on high-quality training data, but it is unclear where such data can be sourced across different domains like education and healthcare. Real-world datasets are often noisy, biased, or incomplete, which can affect the reliability of the model’s predictions. Additionally, LLMs inherently learn statistical patterns at the population level, making them effective for broad generalizations but less suited for capturing individual-specific nuances due to the lack of explicit memory or personalization.

The framework is evaluated only on Q&A-style tasks such as OpinionQA, student assessment, and the Twenty Questions game, which focus primarily on eliciting missing information rather than complex decision-making. This raises concerns about how well the method generalizes to more challenging tasks, such as medical diagnosis, scientific research, or multi-step reasoning, where information acquisition needs to be integrated with reasoning and planning.

The approach requires multiple forward passes through the LLM to compute uncertainty via response sampling, which is computationally expensive. This makes it impractical for real-time applications, such as interactive tutoring or live medical diagnostics, where response speed is crucial. Reducing reliance on repeated sampling, optimizing inference efficiency, or leveraging smaller models for uncertainty estimation could help mitigate this issue.

**Methods And Evaluation Criteria:**

Since the framework does not explicitly define or model the latent variable U, it remains unclear what the model has actually learned about the underlying information structure. This lack of interpretability makes it difficult to trust the system’s reasoning process, especially in high-stakes applications like education assessment or medical diagnosis. A more structured approach, such as probabilistic graphical models (e.g., Variational Autoencoders, Hidden Markov Models), could improve transparency.

The paper primarily uses PPL as the main measure of uncertainty, but PPL only reflects how well a language model predicts text sequences, not the true epistemic uncertainty about the latent variable. In scenarios where the dataset is imbalanced, the model may simply reinforce dominant patterns—for example, assuming that all patients have high blood pressure because it is the most common label in the data. This can lead to overconfident yet incorrect predictions, which undermines the reliability of uncertainty-based question selection.

**Other Comments Or Suggestions:**

See Methods And Evaluation Criteria and Experimental Designs Or Analyses.

**Other Strengths And Weaknesses:**

See Methods And Evaluation Criteria and Experimental Designs Or Analyses.

**Questions For Authors:**

See Methods And Evaluation Criteria and Experimental Designs Or Analyses.

**Relation To Broader Scientific Literature:**

The use of perplexity (PPL) to quantify uncertainty relates to prior work on uncertainty-aware machine learning, particularly in Bayesian deep learning and active learning. Traditional methods, such as Monte Carlo Dropout (Gal & Ghahramani, 2016) and Deep Ensembles (Lakshminarayanan et al., 2017), explicitly model uncertainty by estimating the variance of predictions. In contrast, this paper proposes a novel use of LLM perplexity as a proxy for predictive uncertainty, which aligns with prior research on entropy-based uncertainty estimation in NLP (e.g., Maaløe et al., 2019). However, unlike standard Bayesian methods, this approach does not explicitly separate aleatoric and epistemic uncertainty, which may limit its robustness.

The idea of selecting optimal questions to reduce uncertainty aligns with Bayesian Experimental Design (Chaloner & Verdinelli, 1995) and active learning methods (Settles, 2010), which aim to iteratively collect informative data. Prior work in educational assessment (Reich, 2012) and personalized learning (Piech et al., 2015) has explored similar strategies for adaptively selecting test questions to estimate student knowledge. This paper extends these concepts by using large language models to dynamically generate and select questions based on expected information gain (EIG). The approach is conceptually similar to Pólya tree priors (Hanson, 2006) in Bayesian adaptive testing but replaces probabilistic models with LLM-driven heuristics.

Recent studies have explored using LLMs as agents for reasoning and interactive decision-making (e.g., Brown et al., 2020 (GPT-3); Wei et al., 2022 (Chain-of-Thought Reasoning)). This paper contributes to this line of research by demonstrating how LLMs can be used for adaptive information gathering, a capability related to human-like cognitive strategies for active inference (Friston, 2010) and rational metareasoning (Griffiths et al., 2019). Unlike previous work on prompting-based reasoning, this study frames LLMs as active participants in information elicitation, rather than passive responders.

The use of Monte Carlo Tree Search (MCTS) for multi-step question selection connects this work to reinforcement learning (Silver et al., 2016, AlphaGo) and Bayesian optimization for decision-making (Snoek et al., 2012). While MCTS has been widely applied to game playing and planning problems, its application to adaptive questioning with LLMs is relatively novel. However, this paper does not fully integrate reinforcement learning techniques, which distinguishes it from related work in LLM self-improvement (Schick et al., 2023).

**Theoretical Claims:**

Correct

---

> ### Author Rebuttal · Authors · 2025-03-31
>
> We thank the reviewer for the time and consideration taken in reviewing our paper, and for recognizing our contribution to the application of uncertainty-driven question selection in important domains like education and healthcare.  Below, we respond to your particular concerns.
>
> **[Q1: Modeling the latent, relationship to perplexity, and separation of epistemic + aleatoric uncertainty]**
>
> Most latent entities have no obvious physical meaning (e.g., academic proficiency or political opinions), and can only be observed through noisy observations (e.g., student answers where they guess if they're unsure).
> Our framework avoids the need to perform such direct modeling, allowing us to build a framework that is able to be applied on existing LLMs to use their internet-scale pretrained knowledge to make robust, adaptive decisions.
>
> Our formulation of uncertainty quantification derives from a large body of work on inference with missing data (e.g. [1-2]), and we will include a description of this relationship and our characterization of
> epistemic and aleatoric uncertainty in the paper.  Under this view, knowing $U$ is defined as being able to predict any answer $Y$, with errors only due to random, aleatoric variation. Observing the infinite sequence $(X_{1:\infty}, Y_{1:\infty})$ reduces all epistemic uncertainty, leaving only aleatoric uncertainty: $H(Y \mid U) = H(Y \mid X_{1:\infty}, Y_{1:\infty})$. Then, epistemic uncertainty in $U$ naturally corresponds to uncertainty in the *missing data* $Y_{t+1:\infty}$ given history $Y_{1:t}$. The joint likelihood $P(Y_{t+1:\infty} \mid X_{1:t}, Y_{1:t})$ exactly quantifies uncertainty about the missing data $Y_{t+1:\infty}$ which shows it is the correct objective. Under the special condition of exchangeability, this is precise: there exists a function $\theta(X_{1:\infty}, Y_{1:\infty})$, such that $H(Y \mid U) = H(Y \mid \theta(X_{1:\infty}, Y_{1:\infty}))$ exactly.
>
> Simulating future trajectories allows us to quantify which actions will reduce the most epistemic uncertainty. To quantify uncertainty about a target $Z$, we calculate
>     $$\text{Epistemic Uncertainty}(X_{1:t}, Y_{1:t}) = (\text{Total}) H(Z \mid X_{1:t}, Y_{1:t}) - (\text{Aleatoric}) \mathbb{E}[H(Z \mid X_{1:\infty}. Y_{1:\infty})]$$ If we could infinitely simulate $Y \sim p_{\theta}$, we can exactly calculate epistemic/aleatoric uncertainty. In practice, we will simulate finite time steps to approximate this quantity.
>
> In response to this concern, we will work to clarify these discussions in our revised paper.
>
> **[Q2: Data requirements]**
>
> We appreciate the reviewer's feedback about issues such as dataset imbalance or insufficient data. We will include a grounded discussion of the fundamental limitations of our approach---also shared by other approaches---in the camera ready version. While LLMs may learn average-case behavior, our meta-learning approach is designed to personalize to individuals by understanding what data to gather about each specific individual. A key advantage of our method is we can include any/all user information for the LLM to condition on for sufficient personalization.
>
> **[Q3: Complex Reasoning]**
>
> Our work takes on a different approach to reasoning by integrating explicit uncertainty quantification and estimation with the natural language capabilities of LLMs. Our method can be complementary to reasoning models such as OpenAI's O1 and DeepSeek-R1 by explicitly integrating reasoning with our uncertainty estimation capabilities. Such ideas are beyond the scope of this work, as we aim to first demonstrate the validity of our UQ framework.
>
> **[Q4: Computational Expense]**
>
> While MCTS can be a bottleneck in settings with large question spaces or strict real-time constraints, it remains practical for many applications including those we study. In education, for example, computation can occur during a student’s response time, causing only minor lag. In contrast, real-time tasks like diagnostics or dialogue may require faster methods such as greedy selection. Our work offers a new perspective on uncertainty-guided questioning, and we hope it inspires more efficient approaches (e.g., via distillation).
>
> We also note that methods like embedding-based methods will require a linear scan of the possible questions, yielding a complexity similar to our greedy method. Furthermore, methods that directly use LLMs to generate questions and responses require forward passes in the number of tokens in each question and responses, while our method only requires one forward pass in this respect. We will include a detailed discussion of complexity/tradeoffs in the camera-ready version.
>
> [1] Edwin Fong, Chris Holmes, and Stephen G Walker. Martingale posterior distributions. Journal of the Royal Statistical Society, Series B, 2023.
>
> [2] Naimeng Ye and Hongseok Namkoong. Exchangeable sequence models quantify uncertainty over latent concepts. arXiv preprint arXiv:2408.03307, 2024.

---

### Decision · Program_Chairs · 2025-05-01

**Decision:**

Accept (poster)

**Comment:**

This paper proposes a new framework for eliciting latent information using specifically fine-tuned LLMs, which can be used to reduce uncertainty in various follow-up applications. The reviewers agree that the studied problem is novel and important, and the proposed approach is well supported. During the rebuttal period, the authors addressed most of the concerns. Overall, I think this paper would be a good addition to the community.